# RNA export through the nuclear pore complex is directional

Asaf Ashkenazy-Titelman[1], Mohammad Khaled Atrash[1], Alon Boocholez[1], Noa Kinor[1] & Yaron Shav-Tal [1]✉

The changes occurring in mRNA organization during nucleo-cytoplasmic transport and export, are not well understood. Moreover, directionality of mRNA passage through the nuclear pore complex (NPC) has not been examined within individual NPCs. Here we find that an mRNP is compact during nucleoplasmic travels compared to a more open structure after transcription and at the nuclear periphery. Compaction levels of nuclear transcripts can be modulated by varying levels of SR proteins and by changing genome organization. Nuclear mRNPs are mostly rod-shaped with distant 5'/3'-ends, although for some, the ends are in proximity. The latter is more abundant in the cytoplasm and can be modified by translation inhibition. mRNAs and lncRNAs exiting the NPC exhibit predominant 5'-first export. In some cases, several adjacent NPCs are engaged in export of the same mRNA suggesting 'gene gating'. Altogether, we show that the mRNP is a flexible structure during travels, with 5'-directionality during export.

During transcription, the pre-mRNA is capped, spliced, and poly-adenylated resulting in a mature and processed mRNA that is bound by many proteins. The mature mRNP, namely, a ribonucleoprotein particle, diffuses through the nucleoplasm until reaching the nuclear periphery where it interacts with the nuclear pore complex (NPC) for export into the cytoplasm. It is suggested that the mRNP can undergo restructuring during its travels through the cell. While many proteins assemble co-transcriptionally, some bind later in the nucleoplasm and might be involved in the structuring and compaction of the mRNP particle during its travels[1–6].

The mRNP is considered to be a globular particle. In a study performed in vitro on *Xenopus* mRNPs using naked TFIIA mRNA and an mRNA-binding protein named FRGY2, the measured length of the mRNP was between 90–150 nm, averaging at 125 nm, with a diameter of 10–20 nm[7]. A different in vitro study, examining the mRNP-YB1 complex made similar observations on 2 different genes. mRNP particles formed globular structures resembling beads on a string. The distance measured between globules was about 25 nm, and the average length of the entire mRNP was ~130 nm[8]. Yeast mRNP particles obtained by pull-down assays of the export factor Nab2, which binds to the polyA tail of certain transcripts, were found by EM to be ribbon-shaped and roughly the same size (~30 nm), even though the genes varied in size[9]. Compared to the length of the transcripts, the particles were 10–15 times more compact. Since Nab2 binds the mRNP in the nucleus, the mRNPs were presumably nuclear. In mammalian cells, interactions between RNA-binding proteins (RBPs) from the exon junction complex (EJC) and SR protein families, implemented by high-throughput techniques to study the interactome, were suggested to govern the compaction and packaging of the nuclear mRNP[4]. Studies in yeast directly showed that in the cytoplasm, transcripts can have a circular shape. The 5' and 3'-ends are suggested to interact via the 5'-cap and the 3'-polyA tail to form a circular structure, mediated by interactions between Pab1p and eIF4E-eIF4G, resulting in the enhancement of translation[10]. Circularization was directly demonstrated in intact mammalian cells by RNA FISH with probes labeling different portions of mRNA transcripts, however surprisingly, in most cases, the 5' and 3'-ends of cytoplasmic transcripts were not found in close proximity[11]. Rather, reduction of ribosome occupancy resulted in compaction and increased 5'–3' proximity. Compaction was also seen when the mRNAs localized to stress granules[11,12]. In the nucleus, mRNPs had an extended conformation[9,11] similar to measurements performed on EJC-associated mRNPs, showing flexible rod-shaped structures[13].

[1]The Mina & Everard Goodman Faculty of Life Sciences & Institute of Nanotechnology, Bar-Ilan University, Ramat Gan 5290002, Israel.
✉e-mail: Yaron.Shav-Tal@biu.ac.il

During export, the mRNP can undergo restructuring to accommodate passage through the NPC channel as observed in fixed and living cells[14]. This begs the question of whether there is directionality in the movement of the mRNA through the NPC. Classic studies have focused on one particular mRNP easily detected by EM, namely, the unique and very large Balbiani ring (BR) mRNP (~50 nm in diameter) expressed in the salivary glands of the dipteran *Chironomus tentans*[15,16]. This mRNP is a highly condensed rounded particle in the nucleus and is bound by SR proteins such that it is ~200-fold more compact than the large linear mRNA within it[17]. Due to its large diameter, during export, the BR mRNP unfolds and becomes ribbon-shaped. These studies inferred that the 5′-end is the first part of the transcript to dock and enter the NPC, to then emerge in the cytoplasm, since ribosome recruitment and cap-binding proteins were detected on the protruding end of the transcript[15,18–20]. Most importantly, canonical mRNA export factors are recruited to the 5′-end of the transcript, altogether implying that 5′-first passage is the mechanism governing export directionality[21]. However, direct evidence demonstrating that in mammalian cells the 5′-end of transcript exits the NPC first has yet to be produced.

Several fundamental questions remain open. It is unclear whether mRNAs undergo significant structural changes from the point of release after transcription, through diffusion through the nucleoplasm, and in preparation for export. Regarding the proximity of the mRNA ends, is close proximity unique to the cytoplasm only? Finally, many of the abovementioned studies were not conducted in mammalian cells due to a lack of appropriate tools, particularly, with respect to the issue of whether there is directionality when the mRNA passes through the NPC. Furthermore, do the rules of mRNA directionally during export also govern the exit of long non-coding RNAs that are not translated?

In this study, we address the compaction states and export directionality of mRNAs and lncRNAs at the single-molecule level. We find that nuclear mRNAs are mostly rod-shaped (distant 5′ and 3′-ends), but surprisingly transcripts with the 5′ and 3′-ends in proximity are detected in the nucleus. Different compaction levels for the same mRNA are found in different regions of the nucleus, and compaction levels can be modulated either by increasing the levels of certain RNA binding proteins or by changing the nuclear environment and space available for mRNA travels. mRNAs and lncRNAs detected during export head out of the nucleus 5′-end first, and for some transcripts, several NPCs in the same area export the same RNA species, suggestive of 'gene gating'.

## Results

### Nuclear localization affects mRNP compaction

Using single-molecule RNA FISH (smFISH) with differently tagged probe sets that can label two or three parts of the same transcript, specifically, the 5′-end, the middle and the 3′-end of a certain transcript, it is possible to obtain information on the conformation of the mRNA studied within the sub-cellular compartments. To reliably distinguish between the different parts of the transcript, five long endogenous RNAs, three mRNAs, and two lncRNAs, were examined. MKI67 mRNA (encodes a proliferation factor, from a ~11.6 kbp transcript), TPR mRNA (encodes an NPC basket protein, ~9.6 kbp), NCOA3 mRNA (encodes a cellular receptor, ~7.9 kbp), TUG1 lncRNA (~7.6 kbp) and NORAD lncRNA (~5.4 kbp). We measured the cytoplasm/nucleus (C/N) ratios of the endogenous transcripts which were found to range between 3 and 7.5 (Fig. 1a). Most mRNAs tend to be cytoplasmic, as expected, as were these transcripts. We began with measurements of mRNAs in the nucleus. Typically, there are few copies of a certain mRNA in the nucleus, and so these transcripts were not suitable for the nuclear measurements. To reliably explore mRNA conformation in the nucleus, we used a long transcript that tends to have many nuclear copies, expressed in a cell line we previously generated to track mRNA

transport and export in living cells[22]. The transcript encodes Dystrophin fused to GFP and contains 24x MS2 sequence repeats in its 3′UTR. This gene was termed GFP-Dys-MS2 (~14 kbp mRNA). GFP-Dys-MS2 transcripts were mainly nuclear, and the C/N ratio was significantly smaller, at ~0.1 (Fig. 1a). In the following figures, the 5′-ends of the transcripts will be represented in magenta and the 3′-ends in green.

We first established the detection of the 5′ and 3′-ends of the long GFP-Dys-MS2 transcripts. mRNAs were labeled with two RNA FISH probe sets, one to the 5′-end (probes hybridizing with the GFP-encoding region) and one to the 3′-end (probes hybridizing with the MS2 repeats). Images were acquired by STED super-resolution microscopy. The GFP-Dystrophin-MS2 gene is under inducible expression control[22], and therefore the levels of expression can be modulated. To follow the emergence of the 5′ and the 3′ signals of the transcript during transcription from the time point of transcription initiation, the gene was induced with Ponasterone A, and cells were fixed after short periods. During this time course, the 5′-end of the transcript was the first to emerge at the active site of transcription within the first 5 min of activation (Fig. 1b and Supplementary Fig. 1). The 3′-end appeared only later and was clearly visible after 10 min. Transcription rates for RNA polymerase II range at 2–4 kbp per minute[23–26], which is in agreement with the kinetics detected for the synthesis of the 14 kbp GFP-Dystrophin-MS2 transcript[22].

We then verified the precision of the STED system in measuring single mRNAs. The MS2 part of the GFP-Dys-MS2 transcript was labeled using probes to the MS2 region in two different colors and established that there was no chromatic shift (Supplementary Fig. 2). We then demonstrated that it is possible to separately detect the different ends of the same transcript and mRNP compaction measurements were performed by measuring the distances of the 5′ and 3′ signals of the GFP-Dys-MS2 transcripts by STED, within three different subregions of the nucleus (Supplementary Fig. 3a, b): (i) In the vicinity of the active transcription site, assuming that these are mostly nascent mRNPs recently released from the gene after transcription; (ii) In the nucleoplasm, where the mRNPs are diffusing; (iii) in the nuclear periphery, where mRNPs can bind to the NPC for export. 5′–3′ distances were measured for hundreds of transcripts (Supplementary Fig. 3c, d) and presented as a histogram, representing the percentages of transcripts that showed different 5′–3′ distances, with respect to their sub-nuclear location (Fig. 1c). The analyzed populations showed different types of organization with relation to the distance between the mRNA ends: nucleoplasmic transcripts were the most condensed, where most of the transcripts measured demonstrated a 5′–3′ distance smaller than 30 nm, and a small fraction (<10%) showed distances of more than 60 nm. Most of the transcripts found close to the site of transcription showed a 5′–3′ distance of 30–60 nm, somewhat larger than the nucleoplasmic transcripts, meaning that the transcription site-adjacent population tends to be less compact and might underscore that full maturation of the mRNP occurs after release from the gene. In the peripheral population, a scattered distribution was observed; roughly the same percentage of transcripts demonstrated 0–30, 30–60, or 60–90 nm distances. Notably, compared to the two other regions, at the nuclear periphery, many transcripts were relatively uncompact. These data might suggest that some RNA binding proteins that are responsible for compaction of the mRNP bind only after transcription and that changes that occur to compaction at the nuclear periphery may be in preparation for export.

### Overexpression of SR proteins affects mRNA compaction

Certain studies have suggested that the binding of RNA-binding proteins (RBPs) to the transcript can have an effect on the degree of mRNA compaction[4,13]. We tested the effects of Serine-Arginine (SR) proteins, which are regulators of splicing and export[27]. A study characterizing the EJC interactome showed which SR proteins tend to bind the mRNA, and importantly, it was found that these RNA-binding proteins bind to

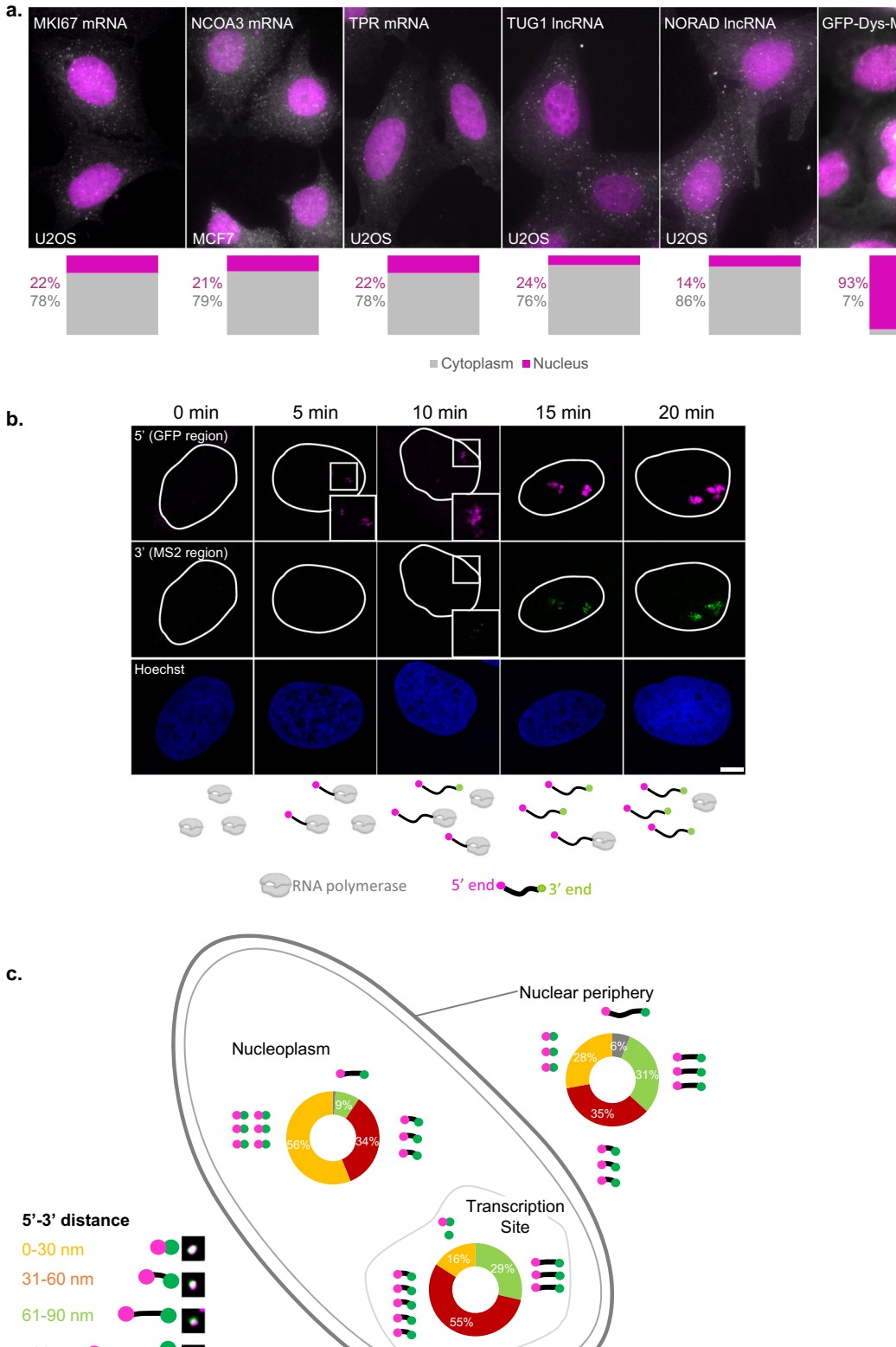

**Fig. 1 | mRNAs change their compaction state during travels through the nucleus. a** Wide-field images showing RNA smFISH on five different mRNA and lncRNA transcripts (white) along with nuclei (magenta) in U2OS and MCF7 cells. Cytoplasm/nucleus (C/N) ratios for each transcript (gray–cytoplasm, magenta–nucleus) are presented below (*n* = 30 cells for each transcript). Scale bar, 10 μm. **b** Confocal images showing activated sites of transcription where the 5′-end (magenta) of GFP-Dys is the first to be transcribed, followed by 3′-end (green). The outlines of the nuclei were traced from the Hoechst images. Boxed areas are enlarged. The scheme at the bottom shows the formation of the transcripts at each time point. The scheme was created with BioRender.com. Scale bar, 8 μm. **c** Summary of the frequency of various 5′–3′ distances measured in the transcripts located in three different regions of the nucleus and the levels of compaction (transcription site, *n* = 147 transcripts; nucleoplasm, *n* = 617 transcripts; nuclear periphery, *n* = 276 transcripts). Each transcript cartoon represents 10% of the total transcripts measured. Examples of images of the various transcripts are shown in the legend. Source data are provided as a Source Data file.

intronless mRNAs as well[4,28–30]. The GFP-Dys-MS2 transcript is intronless. Previously, we showed that the levels of SR proteins in the nucleoplasm are buffered and that modulating their availability and levels in the nucleoplasm can affect gene expression efficiency[31]. We now tested whether the abundance of two SR proteins, SRSF1 and SRSF4, can affect mRNA compaction. According to Singh et al.[4], many copies of SRSF1 were bound to the transcripts they analyzed. On the other hand, SRSF4 had a lower binding efficiency and was barely found on the transcripts analyzed. Therefore, we overexpressed Cerulean-SRSF1 or Cerulean-SRSF4 to increase the levels of these proteins in the nucleus, hence, to examine if they can affect the level of compaction of the transcripts (Fig. 2a). The influence on mRNP compaction was compared to the expression of a Cerulean protein that was not fused to an RBP. While Cerulean had no effect on mRNP compaction, cells that expressed Cer-SRSF1 demonstrated transcripts that were much more compact (Fig. 2b). Interestingly, the greatest effect was on the transcript population that was adjacent to the transcription site, which might indicate that co-transcriptional recruitment of SRSF1 to the mRNA is increased when the RBP is overexpressed. As for SRSF4, only a minor reduction in 5′−3′ distances compared to the Cerulean control was observed, and these were not significant as the effect of SRSF1 (Fig. 2b). This suggests that some RBPs, such as SRSF1, have a compacting effect on the mRNP that takes place probably during transcription and that the degree of compaction depends on the availability of the proteins to bind the mRNA. In addition, it is evident that not all RBPs, and not even all SR proteins, have the same effect on compaction.

To identify the part of the protein that was mediating the compaction effect, a truncated version of SRSF1, termed SRSF1-no-RS, was used, in which SRSF1 was lacking the RS domain, a region of the protein that has repeats of arginine and serine amino acids and can undergo hyperphosphorylation. The latter is important for the splicing activity of the factors via protein-protein interactions and for modulating binding to the mRNA[32,33]. We have shown that this truncated version affects various functions of SRSF1 and the efficiency of splicing[31]. We show that in cells expressing Cer-SRSF1-no-RS there was no significant difference in mRNP compaction, compared to the Cerulean control (Fig. 2b). The data suggest that the RS domain of SRSF1 modulates the effect on the protein's interaction with the mRNP, leading to conformational changes.

## The transcript ends are distant for most but not all nucleoplasmic mRNAs

Using RNA FISH with different probe sets that hybridized with the GFP and MS2 regions of the GFP-Dys-MS2 transcripts (start-to-end, 14 kbp), we could obtain the maximum distances between the 5′ and 3′ ends of the transcript, allowing us to estimate the size of the mRNP within cells. In addition, the dystrophin coding region of the mRNA could also be detected. Using a probe set to the dystrophin coding region combined with either of the GFP and MS2 probe sets, more data about the organization of transcripts in the nucleus was obtained, since the dystrophin probes hybridized closer to the 5′-end than to the 3′ MS2 region. The probe sets were applied to GFP-Dys-MS2 cells activated to express GFP-Dys-MS2 mRNAs (Supplementary Fig. 4a–c). Compared to the labeling of GFP and MS2 regions (5′ to 3′-end, full transcript length) in which the magenta (570 nm) and green (488 nm) signals were separated, the GFP and Dys (5′-end to middle) combination yielded a more overlapping signal, and the same was observed for the labeling of the Dys and MS2 regions (middle to 3′-end) labeling. These distances compared to the start-to-end of transcript measurements (Fig. 3a) showed that the median 5′-end to middle distance was smaller than the middle to 3′-end distance, as the Dys probe binding sites were closer to the 5′-end. Importantly, the 5′-to-3′-end distance was always the largest, compared to the middle to either end, and the 5′-middle distance was always the shortest. This demonstrates the accuracy of the

measurement system. The findings suggest that the nuclear mRNA transcripts are mostly rod-shaped such that the 5′ and 3′ ends are distanced from each other.

Next, we characterized the organization of an endogenous mRNA. To detect mRNAs in the nucleus, which are typically not very abundant, we looked for a long transcript that encodes high levels of mRNA and protein throughout the cell cycle, such that a substantial nuclear mRNA population would be detectable. The *MKI* gene encodes the highly abundant Ki-67 proliferation marker protein and MKI mRNAs were abundant in the nucleus. Using RNA FLAP-FISH[34] in three colors, we examined MKI67 transcripts in U2OS cells. The 5′-end is presented in magenta (Quasar 570 dye), the middle of the transcript in cyan (Cy5), and the 3′-end in green (Fluorescein) (Fig. 3b). Images were acquired by confocal microscopy allowing the combined use of more fluorophores. Nucleoplasmic MKI67 transcripts in U2OS cells were examined and showed that ~75% of the transcripts measured were rod-shaped (Fig. 3c), reaffirming the data obtained from the GFP-Dys-MS2 transcripts. However, for ~25% of the nuclear transcripts measured, the 5′ and 3′-ends were in close proximity with the middle part at a separate distance between them (Supplementary Fig. 4d). This might indicate that nuclear transcripts can undertake different shapes. In contrast, for cytoplasmic MKI67 transcripts (Fig. 3d, e) the major form detected for transcripts tagged with three sets of probes was a distinct triangle-like signal, where the 5′ and 3′ ends were adjacent, and distant from the middle part of the transcript. The two ends of transcript were in close proximity. The analysis suggested that for ~80% of the cytoplasmic MKI67 mRNAs measured, the transcript ends were in proximity while the rest were rod-shaped.

Former work has demonstrated that nucleoplasmic lamin A protein is required for maintaining global genome organization and constrained chromosome movement, while in cells lacking lamin A, the chromatin is loosely organized, dynamic and less compact[35]. We hypothesized that such conditions reducing the interchromatin space in which mRNAs diffuse[22] would affect the compaction levels of the mRNAs in the nucleoplasm. The 5′-end to middle distances of endogenous nuclear MKI67 transcripts were compared in U2OS cells versus U2OS cells in which lamin A was knocked-out, and showed that indeed the mRNAs in the nuclei of KO cells were more compact (Supplementary Fig. 5a–d). We then reversed genome organization to induce chromatin compaction by applying a hyper-osmolar environment to the cells for 10 min[22,36], and the compaction levels of the nucleoplasmic mRNAs reverted to regular levels (Supplementary Fig. 5c, d). These results suggest that the nuclear environment and DNA organization can affect mRNA organization within the mRNP.

## ATP depletion and translation inhibition affect the organization of cytoplasmic mRNA

We tested whether we could reduce the population of mRNAs with transcript ends in close proximity, to shift them to an elongated form. The proximity between the transcript ends has been related to efficient translation activity[10]. Translation is a high energy-consuming process, especially in cancer cells, that can be affected by ATP depletion[37,38]. Cellular ATP levels were perturbed using two different agents: 2-deoxy-D-glucose, a glucose analog inhibiting glycolysis, provoking GDP accumulation in the cell[39]; and Na-azide, an oxidase inhibitor causing mitochondrial malfunction[40] (Fig. 4a). Even 15 min of 2-deoxy-D-glucose treatment were sufficient for inducing a significant reduction in the degree of proximity between the ends of the cytoplasmic mRNAs. Specifically, while 84% of the MKI67 mRNAs had ends in proximity in untreated cells, these percentages were reduced to ~60% in treated cells (Fig. 4b). The percentages did not significantly change when the two ATP depletion techniques were combined, and cells were subjected to the treatment for a full hour. Namely, ~60% of the MKI transcripts were unaffected by the reduced ATP levels. Next, we directly inhibited translation elongation by Cycloheximide[41], given that

a.

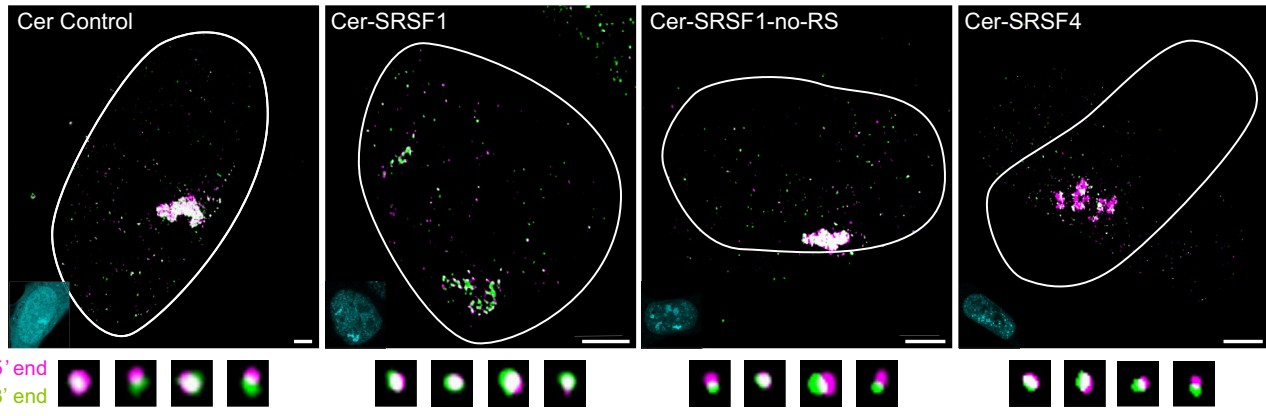

b.

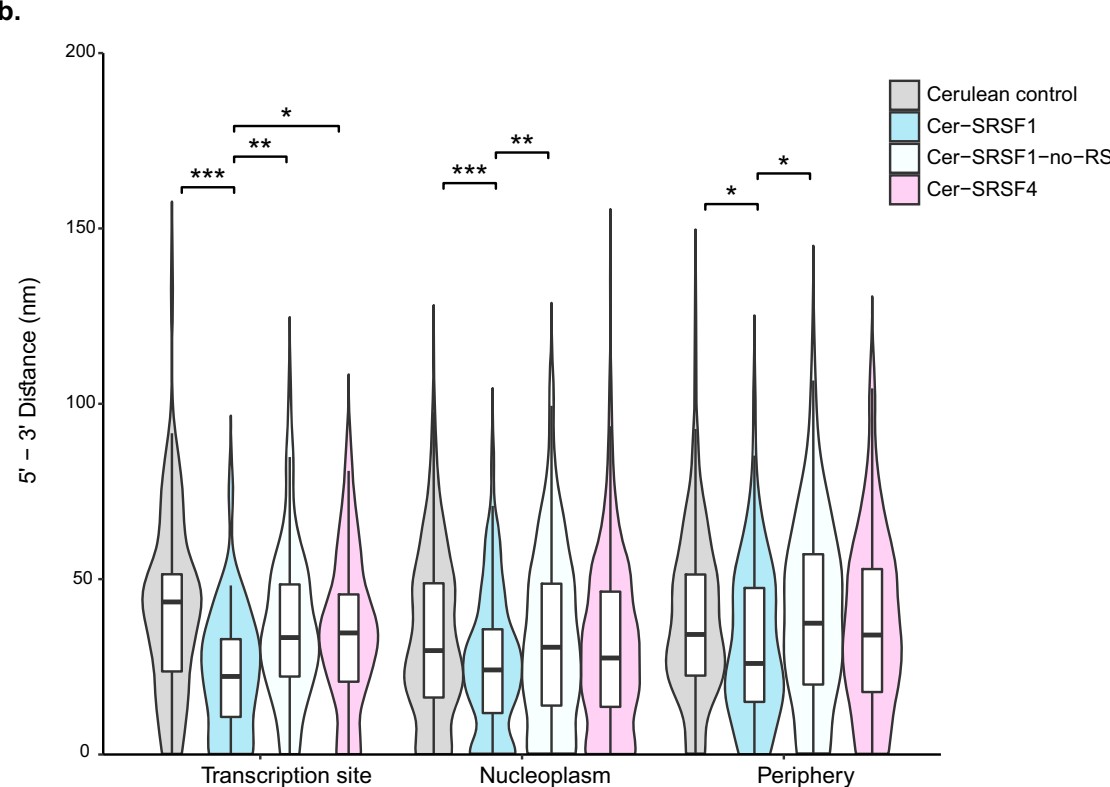

**Fig. 2 | The effects of SR overexpression on mRNA compaction. a** Representative cells expressing GFP-Dys-MS2 transcripts transfected with Cerulean-N1, Cer-SRSF1, Cer-SRSF1-no-RS, or Cer-SRSF4. Large foci are the active transcription sites and the small dots are the mRNAs. The outlines of the nuclei were traced from the Hoechst images. Fluorescent images of the nuclei appear at the bottom left (cyan). Examples of various transcripts are shown below. Scale bar, 5 μm. Adjustments to individual color channels were made. **b** Violin plots depicting changes in mRNA compaction (5' to 3' distance measurements) under overexpression of different SR proteins. Cerulean control $n = 720$ transcripts, Cer-SRSF1 $n = 393$ transcripts, Cer-SRSF4 $n = 508$ transcripts, Cer-SRSF1-no-RS $n = 508$ transcripts. (Transcription site: Kruskal–Wallis, $\chi^2(3) = 17.531$, $p = 0.0005$. Pairwise comparisons- two-sided Mann–Whitney tests with Benjamini–Hochberg FDR correction, Cerulean control vs. Cer-SRSF1 $p = 0.0003$, Cerulean control vs. Cer-SRSF1-no-RS $p = 0.1491$, Cerulean control vs. Cer-SRSF4 $p = 0.1491$, Cer-SRSF1 vs. Cer-SRSF1-no-RS $p = 0.0094$, Cer-SRSF1 vs. Cer-SRSF4 $p = 0.0168$, Cer-SRSF1-no-RS vs. Cer-SRSF4 $p = 0.9103$. Nucleoplasm: Kruskal–Wallis, $\chi^2(3) = 16.374$, $p = 0.0009$. Pairwise comparisons-two-sided Mann–Whitney tests with Benjamini–Hochberg FDR correction, Cerulean control vs. Cer-SRSF1 $p = 0.0005$, Cerulean control vs. Cer-SRSF1-no-RS $p = 0.6236$, Cerulean control vs. Cer-SRSF4 $p = 0.1289$, Cer-SRSF1 vs. Cer-SRSF1-no-RS $p = 0.007$, Cer-SRSF1 vs. Cer-SRSF4 $p = 0.0594$, Cer-SRSF1-no-RS vs. Cer-SRSF4 $p = 0.3742$. Periphery: Kruskal–Wallis, $\chi^2(3) = 8.2911$, $p = 0.0403$. Pairwise comparisons- two-sided Mann–Whitney tests with Benjamini–Hochberg FDR correction, Cerulean control vs. Cer-SRSF1 $p = 0.032$, Cerulean control vs. Cer-SRSF1-no-RS $p = 0.5516$, Cerulean control vs. Cer-SRSF4 $p = 0.5516$, Cer-SRSF1 vs. Cer-SRSF1-no-RS $p = 0.032$, Cer-SRSF1 vs. Cer-SRSF4 $p = 0.2201$, Cer-SRSF1-no-RS vs. Cer-SRSF4 $p = 0.4191$). *$p < 0.05$, **$p < 0.005$, ***$p < 0.001$. Boxplots show the distance (nm) (center line, median; box limits, upper and lower quartiles; whiskers, 1.5× interquartile range). Source data are provided as a Source Data file.

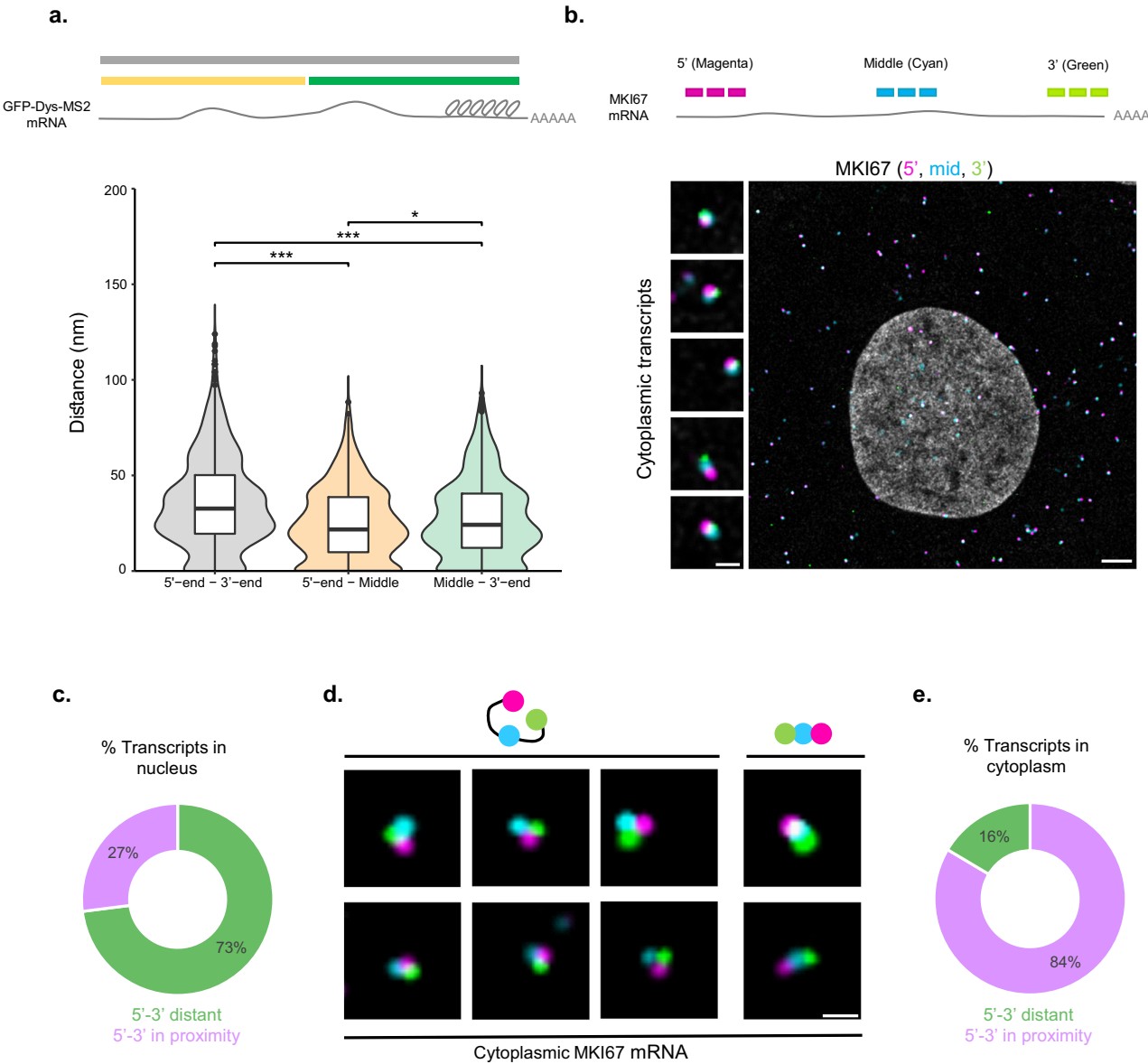

**Fig. 3 | Nucleoplasmic mRNAs are mostly rod-shaped. a** Scheme of the GFP-Dys-MS2 mRNA and the distances measured. Violin plot depicting start-to-end, start-to-middle and middle-to-end distances at three different nuclear regions. 5′–3′ $n = 1040$; 5′-mid $n = 746$; mid-3′ $n = 917$ transcripts. (Kruskal–Wallis, $\chi^2(2) = 12.825$, $p < 0.0001$. Pairwise comparisons–two-sided Mann–Whitney tests with Benjamini–Hochberg FDR correction, 5′-end-3′-end vs. 5′-end-Middle $p < 0.0001$, 5′-end – 3-end vs. Middle-3′-end $p < 0.0001$, 5′-end-Middle vs. Middle-3′e-nd $p = 0.013$). *$p < 0.05$, ***$p < 0.001$. Boxplots show the distance (nm) (center line, median; box limits, upper and lower quartiles; whiskers, 1.5× interquartile range; dots, outliers). **b** Scheme showing the sites of recognition between probe sets and the different mRNAs. MKI67 mRNA in U2OS cells, tagged with three sets of probes (5′−magenta, middle−cyan, 3′− green). Left- enlargements of example mRNAs demonstrating several orientations of the mRNA. Scale bar, 5 μm. **c** Plot showing the shape of nuclear MKI67 transcripts in U2OS cells (5′ and 3′-ends are separate), $n = 100$ transcripts. **d** Triple-tagged single mRNA transcripts in the U2OS cytoplasm (5′− magenta, middle−cyan, 3′−green). **e** A plot showing the shape of cytoplasmic MKI67 transcripts in U2OS cells (5′ and 3′-ends are adjacent), $n = 158$ transcripts. Adjustments to individual color channels were made. Source data are provided as a Source Data file.

treatment with translation inhibiting agents was previously shown to have an effect on mRNP organization[11,12]. Applying Cycloheximide for 4 h had a greater effect on the cytoplasmic MKI mRNA transcripts; only 50% of the transcripts had proximal ends. Combination of all treatments (2-deoxy-D-glucose and Na-azide for 1 h, Cycloheximide for 4 h) did not have a synergic effect; under these conditions, the majority of measured cytoplasmic transcripts were rod-shaped, and only 48% had ends in close proximity (Fig. 4b). Finally, we examined the reversibility of this change in compaction. Cells were treated with combined ATP depletion and translation inhibition as above and subsequently washed in a growth medium for 1 h. A substantial increase in the percentages of MKI67 mRNAs with adjacent 5′ and 3′-ends from 48% pre-

wash to 67% after the wash, was observed (Fig. 4b). Next, we compared the compaction of cytoplasmic MKI67 transcripts during premature termination of translation using Puromycin (Supplementary Fig. 6). Both translation inhibition treatments caused enhanced mRNP compaction, as the median distance between the 5′-end and the middle of the transcripts decreased upon Puromycin treatment and Cycloheximide treatments. As above, the enhanced compaction was reversible after a 1 h wash. These effects on mRNA compaction were also observed for a different transcript, NCOA3 (Supplementary Fig. 7). The half-life of an MKI67 transcript is around 6 h[42], suggesting that the changes were observed in existing transcripts, and not from newly generated ones. These findings in turn show that mRNA organization

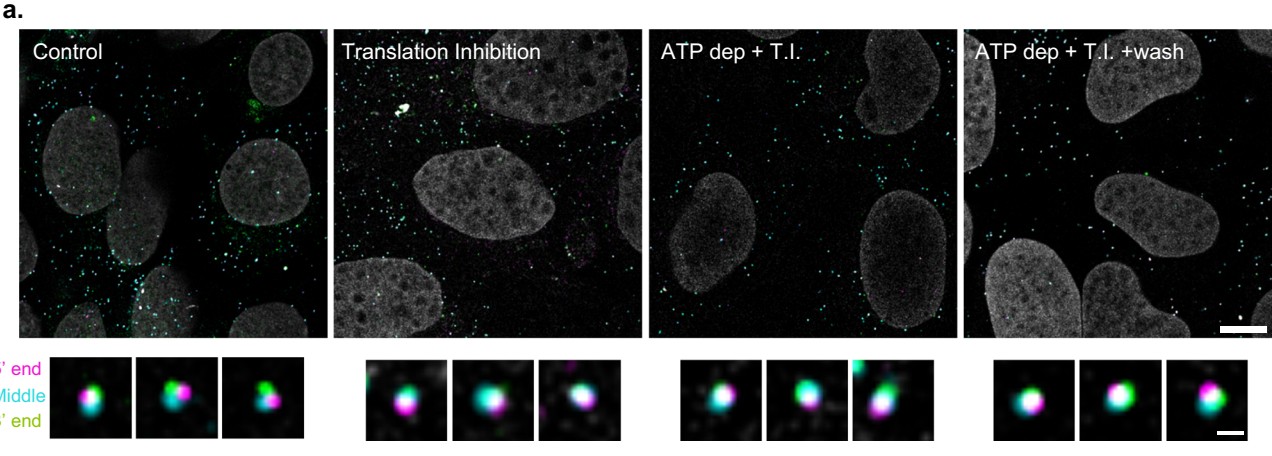

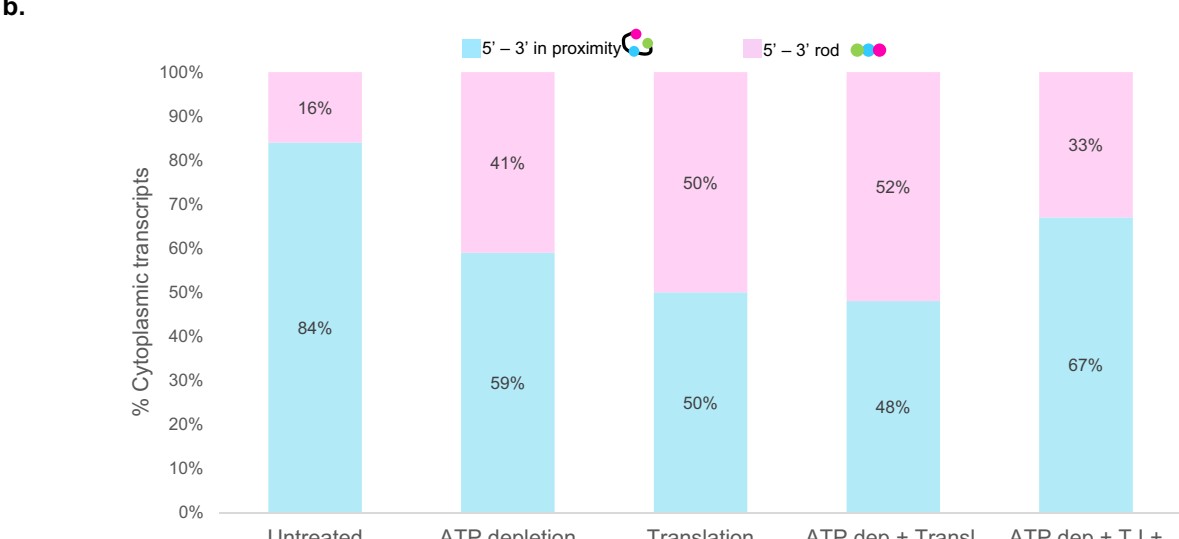

**Fig. 4 | ATP levels and translation state affect mRNA compaction in a reversible manner. a** MKI67 transcripts in U2OS cells, tagged in the 5' (magenta), middle (cyan) and 3'-end (green) of the transcript, along with images of single transcripts (enhanced). Scale bars, 10, 0.5 μm. Adjustments to individual color channels were made. **b** Histograms depicting the percentages of the orientation between the 5' and 3'-ends of triple-tagged MKI67 transcripts in the U2OS cytoplasm, under various treatments. Untreated $n = 158$, ATP depletion $n = 124$, translation inhibition $n = 100$, translation inhibition + ATP depletion $n = 131$, translation inhibition + ATP depletion + wash $n = 143$ transcripts. Source data are provided as a Source Data file.

and compaction are flexible and are responding to the cellular environment, rather than being a one-way process of organization.

### mRNA and lncRNA exit the pore in a 5'-first manner

The 5' and 3'-ends of the MKI67 mRNAs were mostly distant in the nucleus but were found in close proximity in the cytoplasm, suggesting that restructuring of the transcript can occur during the course of transport. Indeed, during mRNA export, the mRNP can undergo restructuring[22,43]. We next examined the directionality of the mRNA and lncRNA ends at the NPC during export. To detect the central region of the NPCs we used POM121 which is a nucleoporin located in the inner NPC region. To verify the correct localization of the expressed protein we transfected the cells with POM121-RFP670. The localization of the POM121 fusion protein in the middle part of the pore structure was verified using confocal and STED microscopy (Supplementary Fig. 8).

Next, mRNA directionality during export was examined using RNA FISH. The NPC was labeled with POM121-Cerulean. Mostly, probe sets in two colors hybridizing at the 5'-end and the middle of the transcripts were used, since the probes marking the 3'-end of the mRNA yielded a relatively poor signal in the NPC, as opposed to a strong signal in the cytoplasm. This way the 5' and middle parts of the transcript could be identified when passing through the NPC. As mentioned, 5 long endogenous transcripts were examined: MKI67 (~11.6 kbp); TPR (~9.6 kbp); NCOA3 (~7.9 kbp); TUG1 (lncRNA, ~7.6 kbp) and NORAD (lncRNA, ~5.4 kbp). MKI67, TPR, TUG1, and NORAD were examined in U2OS cells while NCOA3 was examined in MCF7 cells. Since POM121 is part of the central region of the NPC, we established that a transcript in which at least one of the ends is colocalized with the POM-121 signal is in the midst of mRNA export. All three mRNA transcripts demonstrated the same pattern: the vast majority (~90%) of the transcripts headed out of the nucleus in a 5'-first manner (Fig. 5a, Supplementary Figs. 9a–c and 11a). Few transcripts were located at the NPC in a mid to 5'-end manner. These might be aberrant transcripts or unsuccessful attempts of export. Interestingly, the lncRNA TUG1 exhibited the same directionality

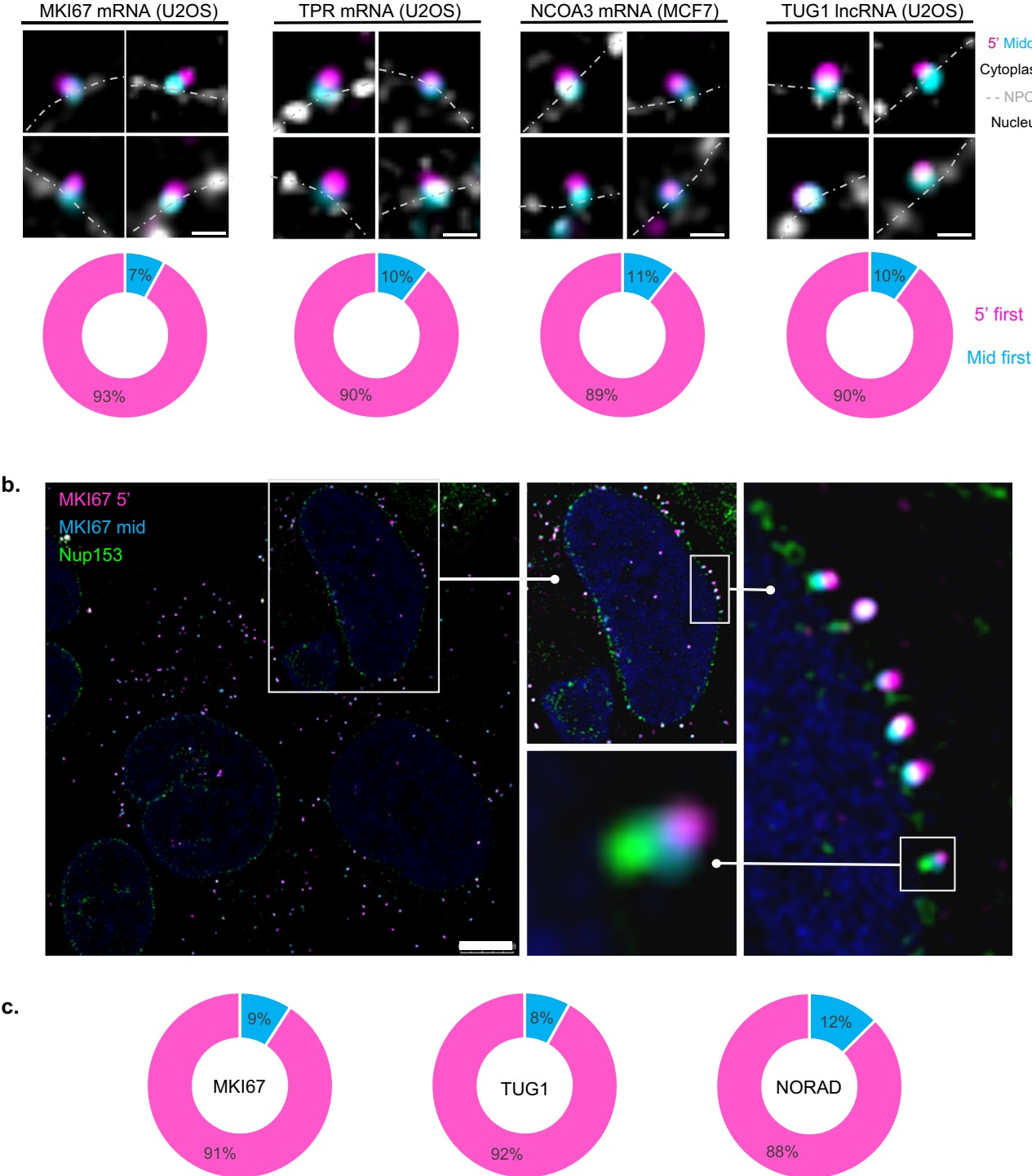

**Fig. 5 | Export directionality of endogenous mRNAs and lncRNAs. a** MKI67, TPR, NCOA3 mRNAs and TUG1 lncRNA double-tagged transcripts in their 5' and middle regions (5'−magenta, middle−cyan) together with POM121-Cer labeling (gray). Charts describing the percentage of transcripts exiting the pore in a 5'-manner (magenta) or middle-first (cyan). MKI67 $n = 105$, TPR $n = 50$, NCOA3 $n = 53$, TUG1 $n = 30$ transcripts. Dotted line−nuclear borders; cytoplasm above and nucleus below the line. Scale bar, 0.5 μm. **b** Double-tagged MKI67 transcripts (5'−magenta, middle−cyan) with endogenous Nup153-mNeonGreen labeling (green). Enlargements are shown in boxes. Scale bar, 8 μm. **c** Charts describing the percentage of transcripts exiting the pore in a 5'-manner (magenta) or middle-first (cyan) in cells expressing Nup153-mNeonGreen. MKI67 $n = 121$, TUG1 $n = 50$, NORAD $n = 40$ transcripts. Adjustments to individual color channels were made. Source data are provided as a Source Data file.

pattern as mRNAs, namely, ~90% of the transcripts exited the pore in a 5'-first manner (Fig. 5a and Supplementary Fig. 9d).

We next generated a U2OS cell line with endogenous tagging of a nuclear basket protein. mNeongreen was knocked into the C-terminus of the *Nup153* gene using CRISPR-Cas12 knock-in (Supplementary Fig. 10)[44]. Nup153 is located at the nuclear part of the NPC where the mRNP binds to it during the initial steps of the export process, and so the 5' and middle parts should be distinct from the basket region as the mRNA moves along the NPC channel. This could be observed, meaning that the transcripts visualized were detected in transit within the pore. Furthermore, the separation between the signals of the two parts of the transcript implies that the mRNA is not globular as it passes through the NPC. Similar percentages (~90%) of measured MKI67 mRNAs as well as TUG1 and NORAD lncRNAs in Nup153-mNeongreen expressing cells, exited the nucleus in a 5'-first manner (Fig. 5b, c and Supplementary Fig. 11b). To determine whether export directionality depends on the translational state, cells were treated with the translation inhibitors, but no change was detected (Supplementary Fig. 12). While for most transcripts identified in association with the NPCs we could find these interactions randomly throughout the whole nuclear periphery, for MKI transcripts, we could identify series of adjacent NPCs in one region of the nuclear envelope that was all exporting the MKI mRNAs (Fig. 5b and Supplementary Fig. 13a) reminiscent of the 'gene gating' model[45]. Since *MKI67* is positioned on chromosome 10 which tends to be in the nuclear periphery, we examined another transcript expressed from a peripheral chromosome. The lncRNA CCAT1 expressed from chromosome 8, also exhibited transcripts that tended to accumulate at NPCs close to the active gene (Supplementary Fig. 13b).

## Discussion

Information about the organization of the mRNA in sub-nuclear areas or the directionality of mRNA/lncRNA export is scarce. We found that the levels of mRNA compaction correlate to the sub-nuclear location in which the mRNA is passing (Fig. 1c). Separate labeling of the 5' and 3' ends using single molecule RNA FISH showed that transcripts in the vicinity of the active gene tended to be quite open compared to the same mRNAs diffusing in the nucleoplasm, that were more condensed. At the nuclear periphery, a heterogeneous population was detected, for which similar percentages of closed, relatively open, or open transcripts, were seen. Peripheral transcripts could be fully mature and export-ready and would suggest that compacted large mRNAs require restructuring prior to export, as the average transcript was more open at the periphery relative to the nucleoplasm.

Nucleoplasmic transcripts would be expected to present a more mature composition of RNA-binding proteins in comparison to the transcripts released from the gene immediately after transcription, which are not fully folded as yet. Indeed, overexpression of SRSF1 that is known to highly associate with mRNAs[4], brought about higher levels of compaction, particularly for the mRNAs close to the active gene (Fig. 2). SRSF1 can interact with the nascent mRNA co-transcriptionally[31,46], suggesting that increasing the abundance of the protein in the nucleoplasm, drove increased co-transcriptional recruitment of these proteins to the pre-mRNA, ending up in a more compact mRNP being released after transcription. This, in turn, would imply that the less compact organization of the mRNA seen when released from the gene under regular conditions, provides the opportunity for post-transcriptional binding of additional RBPs and export factors as the mRNA is traveling through the nucleoplasm. For instance, studies have demonstrated different timings for recruitment of the mRNA export factor NXF1/Tap to mRNA, either co-transcriptionally or post-transcriptionally[30,47,48]. For the gene we were studying, we could not detect NXF1 recruitment to the active gene, which could suggest that in the case of this mRNA, the export factor binds later in the nucleoplasm. The binding of RBPs that affects

compaction is protein specific since overexpression of a different member of the SR protein family, SRSF4, did not have any effect on the compaction levels. An SRSF1 mutant version, lacking the RS domain did not modify the compaction state either. The RS domain of SR proteins is hyperphosphorylated and instrumental for proper splicing[49], and factors lacking this domain lose RNA binding ability, hence, have no effect on mRNP compaction. Altogether, this analysis may imply that increased mRNA compaction is driven by the accumulative binding of RBPs, which also will be key for the diffusional properties of the mRNP traveling in the nucleoplasm[50]. Since diffusion is affected by the diameter of the particle, the more compact the particle, the more rapid its egress from the nucleus would be. The fact that compaction was enhanced in all parts of the nucleus when SRSF1 was overexpressed demonstrates that the location reflects protein binding events, perhaps in correlation to the maturation stage of the transcript. Nuclear mRNP compaction was also affected by the state of genome organization and compaction (Supplementary Fig. 5). We hypothesized that modulating the available space for mRNP diffusion would in turn change the degree of mRNA compaction. Nucleoplasmic lamin A has been shown to uphold chromosome organization and constrain genome mobility, while lamin A knockout cells exhibit increased genome mobility and less compact genome organization[35]. Indeed, in the lamin A knockout cells, where DNA organization is loose and so less interchromatin space is available for diffusion, the transcripts were more compact. Reversing genome organization using hyperosmolar conditions[14,36,50], returned mRNA compaction levels to normal in the knockout cells, demonstrating that the subnuclear environment and available space for mRNA transport also affect the organization of the mRNA within the mRNP (Supplementary Fig. 5c).

The GFP-Dys-MS2 mRNA examined in the nucleus showed that the 5'-middle distance was always shorter on average than the 5'–3' distance, demonstrating that the structure of nuclear mRNP transcripts is rod-shaped with extended ends (Supplementary Fig. 4)[9,11,13]. Examination of the MKI67 endogenous mRNAs showed that ~80% of the nuclear transcripts measured were rod-shaped (Fig. 3). Notably, there was a distinct mRNA population in the nucleus in which the transcript ends were in close proximity. This could infer that nuclear mRNAs can exhibit more than one distinct shape, due to the protein interactome. Proximal 5'- and 3'-ends could imply a circular form for the mRNA. Occurring in the nucleus would imply that circularity is not necessarily an indicator of translation, as has been previously concluded by RNA FISH studies performed on cytoplasmic transcripts[11,12]. Yet, atomic force microscopy and biochemical studies have distinctly shown cytoplasmic transcripts to be circular[10]. The vast majority of cytoplasmic MKI67 transcripts detected by triple-labeling exhibited proximal ends, in which each region of the transcript was spatially distinct and not necessarily overlapping with the other region (Fig. 3d, e). Our findings of proximal transcripts ends in the cytoplasmic transcripts partially concur with some of the conclusions of the previous studies[11], showing that cytoplasmic mRNAs are mainly rod-shaped under regular conditions and that translation inhibition leads to compaction, probably due to the release of ribosomes from the transcripts. Differences between studies might be due to different types of transcripts or the measurement techniques applied.

To probe if the association between the MKI67 transcript ends could be modulated by the translational status of the cell, we perturbed translation conditions (Fig. 4). ATP depletion was sufficient in inducing an organizational change in cytoplasmic mRNAs, occurring within the span of minutes. Still, more than 50% of the transcripts remained with associated ends. The translation inhibitor Cyclohex-imide which inhibits translation elongation caused a larger decrease in association between the transcript ends. In fact, both Puromycin and Cycloheximide prompted mRNP collapse, demonstrating a very compact state of the transcript that was reversible upon washing (Supplementary Fig. 6). We show that this result is not unique to the MKI67

transcript, as measurements done with NCOA3 mRNA (Supplementary Fig. 7) show the same collapse under translation inhibition conditions. These results correlate with the work of Adivarahan et al., which demonstrated enhanced compaction of the MDN1 transcript upon Puromycin treatment, and a slighter change in compaction under Cycloheximide treatment[11]. A small synergistic effect was observed when ATP depletion and cycloheximide were combined, and this effect was reversible by washing out the inhibitors. Given that the half-life of the MKI67 transcript is ~6 h[42], we can assume that most of the transcripts measured after the wash were not newly synthesized. Importantly, even under these conditions that block translation, a significant population remained with associated ends, strengthening the notion that a circular shape for an mRNA is not an indicator for ongoing translation, but rather one of the regulatory steps towards it[11,12]. In addition, the rod-shaped state of the transcripts in an ATP-depleted environment may suggest that reaching a circularized shape, namely bringing the end to close proximity, requires energy, and that the relaxed state of an mRNP is rod-shaped. A previous study has shown that the compaction degree of mRNA was unchanged under conditions of ATP depletion by 2DG together with puromycin to release the ribosomes and that the ability to undergo mRNA compaction was independent of energy[12]. Taken together, this would suggest that energy is not required for compaction, but that lack of energy can lead to changes in mRNA organization.

Former mRNA export studies were based on a cell system in which a single type of a very large mRNA in *C. tentans* could be observed and detected during different stages of transport or export[15,16,18]. The appearance of particles on the portion of the mRNA protruding from the NPC into the cytoplasm suggested that these were ribosomes, which in turn bind the 5′-end of the mRNA. This implied that the 5′-end of the BR mRNA is the first to enter the NPC. In addition, the large mRNP unfolded and progressed 'worm-like' through the NPC. Restructuring of a large mRNP during export was seen in living cells[22]. Recently, it was shown that the width of the NPC channel is flexible[51,52]. Interestingly, a study on the structure of nanoparticles has shown that rod-shaped or elongated nanoparticles were more efficient in their delivery through the NPC compared to spherical nanoparticles[53]. By examining several endogenous mRNAs by RNA FISH in mammalian cells together with the detection of the NPC by POM121 that is situated in the center of the NPC, we showed that ~90% of the transcripts detected at the NPCs exited the nucleus in a 5′-first manner (Fig. 5). Interestingly, the same pattern applied for lncRNAs, even though they are not translated, implying that the 5′-first rule applies to the RNA independent of translation potential (Fig. 5). Indeed, inhibition of translation had no effect on directionality (Supplementary Fig. 12). To obtain a higher resolution of the passage of the different parts of the mRNA within the NPC, we endogenously fluorescently tagged a component of the nuclear basket, Nup153. This allowed the clear detection of the basket via the fluorescent protein, and just above the basket we could detect the middle part of the mRNA which would be localized in the central channel of the NPC, and above that, the 5′ part of the mRNA, demonstrating an open structure of the mRNA and its 'worm-like' passage through the NPC. A very small population of the transcripts seemed to exhibit a middle-first position in the vicinity of the pore. We postulate that either these transcripts might be going through an aberrant export attempt or that folding of the transcript is causing this phenotype.

Interestingly, some cells featured several MKI67 transcripts exiting through pores in the same region of the nucleus. This is rather unusual since for most transcripts we identified randomly positioned interactions with NPCs throughout the nuclear periphery, and studies in living mammalian cells have shown that mRNAs released from a gene close to the periphery diffuse throughout the whole nucleus and do not necessarily exit close to their site of transcription[50,54,55]. It was not uncommon to find 2–4 (in some cases as many as 7) transcripts

exiting pores that are in close proximity (Supplementary Fig. 13a). MKI transcript levels are relatively high in cells, suggesting that the *MKI* gene is frequently transcribing. *MKI* is positioned on chromosome 10 tends to be peripheral and so it is possible that several NPCs in close proximity to the gene would be engaged in MKI mRNA export. A similar observation was seen for a lncRNA transcribed from chromosome 8 which is also a peripheral chromosome (Supplementary Fig. 13b). This is reminiscent of the 'gene gating' hypothesis by Gunther Blobel, fostering the idea that some active genes would be positioned in the vicinity of NPCs, and thus the export of their transcripts would be through designated pores[45].

## Methods

### Tissue culture and transfections

Human U2OS cells were maintained in low glucose DMEM (Biological Industries, Beit-Haemek, Israel), supplemented with 10% fetal bovine serum (HyClone Laboratories, Logan, UT), 1% Glutamine (Biological Industries, Beit Haemek, Israel), and 1% penicillin–streptomycin solution (Biological Industries). MCF7 and HeLa cells were maintained in high glucose DMEM (Biological Industries), supplemented with 10% fetal bovine serum and 1% penicillin–streptomycin solution. All cells were grown at 37 °C in 5% $CO_2$. The GFP-dystrophin-MS2 U2OS cell line[22] was selected using 165 µg/µl hygromycin (Sigma-Aldrich) and activated using Ponasterone A (PonA) at 5 µg/µl for 24 h (unless otherwise specified). Hyper-osmolarity treatment was performed with 1:100 10XPBS for 10 min. ATP depletion was performed using either 2-deoxy-D-glucose (20 mM, Sigma) or Na-azide (20 mM, Sigma) in medium for either 15 min or 1 h. Hoechst 33342 (2.5 µM) was used for labeling DNA. Puromycin (Invivogen) was used at a concentration of 10 µg/ml for 30 min in medium. Cycloheximide (Sigma-Aldrich) was used at a concentration of 100 µg/ml for 4 h in medium.

### Plasmids, transfections, and gene editing

To generate POM121-Cerulean, the Cerulean coding region was inserted (BamHI and EcoRI) into a POM121-RFP670 plasmid[22] replacing the RFP670 coding region. Cer-SRSF1, Cer-SRSF4, Cer-SRSF1-no-RS were previously described[31]. Plasmids were transiently transfected into U2OS cells with the PolyJet transfection reagent (SigmaGen Laboratories).

For CRISPR-Cas9 knockout of the *LMNA* gene, sgRNA sequences were designed using www.benchling.com, with genome version HGCr38. sgRNA sequences (aa5−top: CCATGGAGACCCGTCCCAG, bottom: CTGGGACGGGGTCTCCATGG) were cloned into the BB-Cas9-2A-GFP plasmid (Addgene, PX458) using BpII (Fermentas), as published (Ann Ran et al.[56]), then transfected into U2OS cells using the PolyJet transfection reagent. Transfected cells were sorted for a GFP-positive signal (405 nm laser for excitation and a 450/40 nm bandpass filter for detection) using a BD FACS Aria III cell sorter. After sorting into 96-well plates (1/3/5 cells in each well), cells were grown in DMEM, 20% FBS, streptomycin, and HEPES. Selected clones were subjected to immunostaining, TIDE sequencing, and Western blotting to verify the lack of lamin A protein. Antibodies used: rabbit anti-tubulin at 1:10,000 (Abcam, ab4074) and rabbit anti-lamin A at 1:1000 (Abcam, ab26300). Scans are provided in the Source data file.

For the knockin of the mNeongreen fluorescent protein into the C-terminus of the endogenous *Nup153* gene, we used the Mammalian PCR-tagging protocol for tagging the C-termini of genes[44]. The pMaCTag-P07 plasmid containing the mNeongreen coding region was used (Addgene #124788). We also added MS2 repeats into the construct for future studies. The online oligo design tool www.pcr-tagging.com[44] was used for searching PAM sites and to design the forward (M1) and reverse (M2) tagging oligos specific for LbCas12a/LbCpf1 (Addgene #69988). M1 and M2 oligos, targeting Nup153, were obtained from IDT.

M1: TTTCACAGGTCAAATGGGAAAAATGTGTTCTCTTCTTCTGGA
ACTTCATTCTCTGGTCGCAAGATAAAGACTGCTGTTAGACGCAGGAA
ATCAGGTGGAGGAGGTAGTG

M2: TCTGAAAGCAGGGCACCAGCTGTTGTTAAAATTGAGTACAA
CACCAATGTGACCTAAAAAACTGCTGTTAGACGCAGGAAAATCTACA
CTTAGTAGAAATTAGCTAGCTGCATCGGTACC

The cassette was amplified with M1 and M2, using pMaCTag-P07-NG-MS2V5 as a template. To generate U2OS Tet-On cells expressing endogenously tagged Nup153-mNeongreen cells, ~10⁶ cells were seeded 24 h before transfection. Plasmids containing LbCas12a/LbCpf1 (Addgene pY016, #69988) and the mNeongreen PCR cassette were electroporated (Bio-Rad) into the cells. 72 h later, the cells were trypsinized and Puromycin selection was added. After 7–14 days, cells were examined for mNeongreen fluorescence and seeded into a 96-well plate for the generation of single clones. After another 1–2 weeks, positive clones were transferred to a 24-well plate for further analysis. Genomic DNA from U2OS Tet-On and the knockin Nup153-mNeonGreen cells was extracted with the Wizard® Genomic DNA Purification Kit (Promega). Clones were genotyped by PCR for homozygous insertion of tags and the PCR product was sent for verification sequencing. For PCR verification of the knockin, primers used were: forward GCGTGTGTGTTAGGTGTAGCAG (Nup153 intron 22), reverse TACGGAACACGTACATCG (mNeongreen in the cassette). To verify that mNeongreen was indeed tagging Nup153, siRNA knockdown was performed siRNAs to Nup153: 5′-GGCUACAAAGAUACUUCAA CAAGAA-3′ (duplex 2, 25 nM), negative control 5′-CGUUAAUCGC GUAUAAUACGCGUA-3′ (TriFECTa siRNA kit; IDT). siRNAs were transfected with Lipofectamine 2000 (Invitrogen). 72 h after siRNA transfection, cells were fixed with 4% PFA and imaged.

## Fluorescence in situ hybridization

**Stellaris protocol.** RNA smFISH experiments with Stellaris (Biosearch Technologies) probes were performed according to their adherent cell protocol. Probe sets used were: FAM-labeled MS2 region (excitation 495/emission 520) or Quasar-570-labeled MS2 (Ex 548/Em 566) containing 3 20-mer probes per MS2 cassette; Quasar-570-labeled GFP region; FAM-labeled GFP region; Quasar-670-labeled YFP/GFP region (Ex 647/Em 670); FAM-labeled MKI67 (3′-end); Quasar-570-labeled MKI67 (5′-end), Quasar-570-labeled Dystrophin coding region and Quasar-570-labeled CCAT1 (Supplementary Data 1). All at a concentration of 12.5 μM per coverslip.

**FLAP-FISH.** We implemented the FLAP-FISH protocol as published[34]. Three sets of FLAP tails were used: X (CACTGAGTCCAGCTCGAAA CTTAGGAGG), Y (AATGCATGTCGACGAGGTCCGAGTGTAA), and Z (CTTATAGGGCATGGATGCTAGAAGCTGG). X-tails were conjugated with either Cy3 (Ex 548/Em 566) or Cy5 (Ex 647/Em 670) fluorophores. Y, Z tails were conjugated with either 6-FAM Fluorescein (Excitation 495/Emission 520) or Cy5 fluorophores. 28 nt tails were added to the 5′-end of each probe designed, the reverse-compliment sequence of the fluorescent tail. After diluting the probes to 100 μM, they were mixed and then diluted to a concentration of 0.833 μM. Next, the protocol described in ref. 34 was used to hybridize the probes with their fluorescent tails. FLAP experiments were performed according to the Stellaris adherent cell protocol. Probe sets used: MKI67 middle region (X), MKI67 3′-end (Y), NCOA3 5′-end (X), NCOA3 3′-end (Y), TPR 5′ (X), TPR 3′-end (Y), TUG1 5′-end (X), TUG1 3′-end (Y), NORAD 5′-end (X), NORAD 3′-end (Z) (Supplementary Data 1).

## Immunofluorescence

For all immunostainings, primary antibodies (Abcam) used were rabbit anti-Nup358 at 1:200 (ab64276); rabbit anti-Nup214 at 1:200 (ab70497); mouse anti-Nup62 at 1:300 (ab96134); rabbit anti-Nup153 at 1:300 (ab171074); rabbit anti-TPR at 1:100 (ab170940); rabbit anti-lamin A at 1:200 (ab26300). Secondary antibodies: the secondary antibodies used were anti-mouse Cy3 (ab97035), anti-rabbit Cy3 (ab6939; Abcam), Alexa Fluor 647 goat anti-mouse (A21235), and Alexa Fluor 594 goat anti-rabbit (A11072; Molecular Probes).

## Fluorescence microscopy

Wide-field fluorescence images were obtained using the Cell^R system based on an Olympus IX81 fully motorized inverted microscope (×60 PlanApo objective, 1.42 NA) fitted with an Orca-AG CCD camera (Hamamatsu) driven by the Cell^R software. Super-resolution imaging was performed on a Leica SP8 inverted microscope equipped with a STED module, a pulsed white-light laser, and gating. The objective used was a STED dedicated ×100 1.4 NA, with Leica immersion oil, at room temperature. The mounting medium was homemade 80% glycerol with *p*-phenylenediamine antifade (Sigma), and the cover glasses were high-precision #1.5 (Thermo Scientific). Dual-color experiments were performed by between-line sequential imaging using the 660-nm depletion laser set at 90% (slider) of 90% laser power for FAM and 50% slider for 570 Quasar. This allowed for more accurate spatial imaging (as opposed to using the 592 nm depletion laser for Alexa Fluor 488, which would necessitate between-frame imaging). Gate settings were 0.3 ns for the Rhodamine Red-x and 2.5 ns for Alexa Fluor 488. Confocal images were acquired using the same microscope. The resulting images were deconvolved with Huygens Professional (Scientific Volume Imaging, Hilversum, The Netherlands) using the CMLE algorithm, with signal-to-noise ratios of 12 and 5 iterations, using the Huygens STED module.

## STED calibration and control

To estimate the precision of the imaging system we labeled the MS2 repeats of the GFP-Dys-MS2 transcripts (3′ end) simultaneously with 2 sets of probes, 520 nm (green) and 570 nm (magenta). Since the probes completely overlap, an ideal system would give us a distance of 0 nm. Over 200 transcripts were analyzed, and measurements read an average deviation of 9 nm, while the median was 0 nm, meaning the majority of transcripts measured overlapped perfectly.

## Data analysis

Directionality analysis of the different regions labeled in one mRNA in conjunction with a nuclear pore was performed by detecting FLAP smFISH signal of the 5′-end (magenta, Cy3) and the middle of transcript (cyan, Cy5) that overlapped with a POM121-Cer signal (gray), in images acquired by confocal microscopy. Only transcripts exhibiting a clear differentiation between the two regions of the transcript were used for analysis. All measured transcripts from all NPCs analyzed were plotted in charts. mRNA compaction analysis of endogenous transcripts using FLAP smFISH was performed by localizing the middle of transcript signal (cyan, Cy5) compared to the 5′/3′ ends (magenta/Cy3 or green/Fluorescein, respectively) in images acquired by confocal microscopy. Only transcripts demonstrating 3′-end signals in which the 3 tags were visibly separable were analyzed. The maximum distance measured was ~300 nm, provided that there were no transcripts adjacent to it so that we can conclude that both labels are on the same transcript. Particles were only picked if their size (a signal of 0.2 microns) and intensity (1500-1000 units, LASX) were within range, and transcription sites were excluded from the analysis. Transcripts that were too close to other transcripts were not analyzed (unable to distinguish), to avoid bias. A transcription site is a strong and distinguishable zone where one cannot differentiate between single transcripts. For compaction analysis of the GFP-Dys-MS2 mRNA, a line was manually drawn across each mRNA. The intensity of both channels (5′ to 3′ or middle to ends) along the lines was measured by LAS X software and outputted as an Excel file representing a single transcript, in images acquired by STED super-resolution microscopy. All STED and confocal images were deconvolved using Huygens Professional (Scientific Volume Imaging).

Finally, all measured distances from all transcripts analyzed for each condition were plotted in violin plots.

Calculations of mRNA numbers and C/N ratio were performed using the Imaris spot function to identify each individual mRNA along with a surface analysis on the Hoechst channel. Each field for analysis consisted of 51 Z-planes; images were acquired by wide-field microscopy and deconvolved using Huygens Essential. Only cells that were distinctly separated from others were analyzed. The algorithm was used to differentiate between nuclear and cytoplasmic transcripts, to calculate the C/N ratio.

### Statistical analysis

The experiments presented were repeated at least three times. Statistical analysis and $p$ values are presented in the figure legends. Source data are provided in this paper. A two-tailed $t$-test was performed in *LMNA* knockout measurements. For SRSF overexpression and 5'-mid/mid-3' measurements, populations were compared using a Kruskal–Wallis rank sum test followed by pairwise Mann–Whitney tests as post hoc analysis with Benjamini–Hochberg $p$-value adjustment (FDR) for multiple testing. Violin plots were generated via R.

Following are the numbers of mRNAs and cells examined in the different experiments:

Fig. 1a: total RNAs analyzed: MKI67, 7270 mRNAs, 31 cells; NCOA3, 10203 mRNAs, 30 cells; TPR, 4330 mRNAs, 30 cells; TUG1, 2928 lncRNAs, 30 cells; NORAD, 5018 lncRNAs, 30 cells; and GFP-Dys-MS2, 5766 mRNAs, 30 cells.

Fig. 1c: $n = 1040$ mRNAs (transcription site, $n = 147$ mRNAs; nucleoplasm, $n = 617$ mRNAs; nuclear periphery, $n = 276$ mRNAs).

Fig. 2b: Cerulean control $n = 720$ mRNAs; Cer-SRSF1 $n = 393$ mRNAs; Cer-SRSF4 $n = 508$ mRNAs; Cer-SRSF1-no-RS $n = 508$ mRNAs.

Fig. 3a: 5'-3' $n = 1040$ mRNAs; 5'-mid $n = 746$ mRNAs; mid-3' $n = 917$ mRNAs.

Fig. 3c: $n = 100$ transcripts.

Fig. 3e: $n = 158$ transcripts.

Fig. 4b: Untreated $n = 158$ mRNAs; ATP depletion $n = 124$ mRNAs; translation inhibition $n = 100$, translation inhibition + ATP depletion $n = 133$ mRNAs; translation inhibition + ATP depletion + wash $n = 143$ mRNAs.

Fig. 5a: MKI67 $n = 105$ mRNAs; TPR $n = 50$ mRNAs; NCOA3 $n = 53$ mRNAs; TUG1 $n = 30$ lncRNAs.

Fig. 5c: MKI67 $n = 121$ mRNAs; TUG1 $n = 50$ lncRNAs; NORAD $n = 40$ lncRNAs.

Fig. S1: 5 min, GFP $n = 46$ transcription sites; 10 min, GFP = 44, MS2 = 44 transcription sites; 15 min, GFP = 41, MS2 = 35 transcription sites; 20 min GFP = 36, MS2 = 39 transcription sites.

Fig. S2 MS2-MS2 $n = 290$ mRNAs.

Fig. S5d: U2OS $n = 171$ mRNAs; *LMNA* knockout (Colony 3) $n = 246$ mRNAs; *LMNA* knockout + hyper-osmolar $n = 206$ mRNAs.

Fig. S6: Untreated $n = 128$ mRNAs; Puromycin $n = 87$ mRNAs; Puromycin + wash $n = 93$ mRNAs; Cycloheximide $n = 94$ mRNAs; Cycloheximide + wash $n = 64$ mRNAs.

Fig. S7: Untreated $n = 135$ mRNAs; Puromycin $n = 139$ mRNAs; Cycloheximide $n = 199$ mRNAs.

Fig. S12: Untreated t $n = 67$ mRNAs; Puromycin $n = 58$ mRNAs; Cycloheximide $n = 65$ mRNAs.

### Reporting summary

Further information on research design is available in the Nature Research Reporting Summary linked to this article.

## Data availability

The datasets generated during and/or analyzed during the current study are available from the corresponding author upon reasonable request. The scans and quantification data generated in this study are provided in the Source Data file. Source data are provided with this paper.

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

## Acknowledgements

We thank Jennifer I.C. Benichou (BIU) for the assistance with the statistical analysis, and Irit Shoval and Avi Jacob from the BIU Imaging Facility. This study was supported by the Israel Science Foundation (1278/18; Y.S.-T.). The authors declare no competing financial interests.

## Author contributions

A.A.-T. designed and performed the experiments, measured and analyzed the data, and wrote the manuscript. M.K.A. performed RNA FISH quantifications. A.B. generated the Nup153-mNG cells. N.K. performed transfections. Y.S.-T. designed the project and wrote the manuscript.

## Competing interests

The authors declare no competing interests.
