## [Peer Review File · Nature Communications]

RNA export through the nuclear pore complex is directionalREVIEWER COMMENTS

Reviewer #1 (Remarks to the Author):

Ashkenazy-Titelman et al investigate mRNP conformation using single molecule FISH in 5' to 3' rainbow mode. They nicely address several outstanding questions of mRNP export, in example if mRNPs change their conformation during nuclear export, and in different places of the cell. They show that nascent RNA is more extended, and that RBPs may render mRNPs more compact. In particular the data presented in Figures 5, S5 and S7 are striking. My compliments!

Quite honestly, I think this is a landmark paper. It is very well suited for Nat Comm and I most enthusiastically support its publication.

Some comments:

In the abstract the authors state that: "It is assumed that at the nuclear pore complex (NPC), the mRNP unfolds and enters 5'-first into the pore." In the introduction the authors nicely underline the notion that the 5' end enters first with the relevant literature. But why is it intuitive that to assume the mRNP has to unfold? The authors state themselves that Balbini ring mRNPs are globular during export. Furthermore, FG-Nups and mRNPs have different biophysical properties and thus likely need to minimize their interaction surface.

Line 116: May the massive MS tagging at the 3' end interfere with how the mRNP enters the NPC?

Fig 1B: You should quantify this data.

Fig. S1D: What exactly is the difference between left middle and right panel? Different positions in the nucleus? This should be labeled more clearly. On a general note, the authors often place explanations into the main text that alternatively could be into the figure legends.

Fig 2C: The plot indicates a rather large spread of the data but in this representation its hard to tell. You may choose violin plots or something that allows a more objective judgement.

Fig. 3C: Can one really conclude circularity? Why exactly? Do you really mean physical circularity? The authors somehow would have to show how this differs from random non-circular conformations, e.g. by comparing to a cleaved or otherwise non-circular transcript. This could be addressed by re-phrasing more accurately. You may also compare to data from the next section of the paper (cycloheximide treatment) to address this. Statistical tests would be good. Also for Figure 4B.

Line 274: I suggest to delete 'towards the nucleoplasmic side', unless you have more clear evidence. Please note that Nup62 is a member of both the Nup214 and Nup62 complexes. You see POM121 between basket and CF Nups. That's good enough.

Line 287: You are probably right, but 1st time claims are tricky and may upset people. Nobody can be aware of all literature that has been published.

Line 302/ 419: 'Gene gating' would mean that it is also transcribed at the nuclear periphery, right? Strictly speaking, you have not shown that. There could be alternative explanations (defined tracks towards the export site, splitting of particles close to NPCs or coincidence which would bring us back to statistical tests). You could down town and say that local clustering of exported transcripts may be indicative of gene gating. Or repeat the assay shown in Figure 1B with that particular mRNA and show that its close by.

Reviewer #2 (Remarks to the Author):

In this manuscript Ashkenazy-Titelman and coworkers report a super-resolution (STED) study by smFISH of mRNAs in different cell-compartments. Image resolution seems much more accurate than in most previously published works. Probes of different colours target the 5' ends, the middle and the 3' ends of the investigated transcripts. They describe changes in compaction of the mRNPs across compartments and most importantly observe that export from the nucleus to the cytoplasm begins preferentially by the 5' end of the transcripts. To conclude, this is an important piece of work that requires some additional controls and details.

MAJOR COMMENTS

- 1) When distances between probes on the same molecule are measured, resolution of probes of different colours is critical. Resolution is one of the most spectacular advance in this study. Thus it should be strongly supported at least supplementary data. In the material and methods section, the authors say they estimate the chromatic distortion by labelling repeated sequences with probes of different colours. The data used to determine a 7 nm average deviation, should appear and detailed in Supplementary figures. How many images were analysed? Show the distribution. Is the chromatic shift homogeneously centred around a median of 0 within the entire observation field?
- 2) Is the 7 nm resolution in x,y? What is the resolution in z?
- 3) How are distances between probes determined? Did you take into account that mRNAs are 3D objects? A vertical rod might appear as a "compact" object.
- 4) Histograms in Figure 3C, 3E and 4B lack statistical analysis of distances (i.e.. box plots) and discussion of the cut-off distance between 5'-3' distant and 5'-3' in proximity classes. How many images cannot be ascribed to one or the other class?
- 5) Show data proving that protein synthesis resumes after removal of cycloheximide (Figure 4).
- 6) Is RNA export orientation (Figure 5) sensitive to protein inhibition (cycloheximide) to address a possible contribution of ribosomes to export orientation? Can you rule out that a scanning
- 7) Other labs using different RNAs made different observations for the impact of protein synthesis on mRNA compaction. Is the discrepancy due to the RNAs investigated? One should probe at least one of the previously investigated mRNAs to discriminate between a technical effect or target effect.

MINOR COMMENTS:

Numbers such as 42.892 nm (Fig S1), 7.128 (methods) are meaningless. 43 nm and 7 are more appropriate.

Reviewer #3 (Remarks to the Author):

mRNA export from the nucleus to the cytoplasm via nuclear pore complexes is a fundamental step in eukaryotic gene expression. How individual mRNAs are packaged for successful export is currently unknown, and a long-standing question in the field has been whether there is an orientational preference as it was suggested by classic EM studies performed by the Daneholt group in *Chironomus tentans*. Ashkenazy-Titelman et al. tackle this important problem using RNA fluorescence in situ hybridization (FISH) to determine the compaction state of individual mRNAs at different stages of the mRNA's life, and how distinct parts of an mRNA (the 5' end, the middle part and the 3' end) are oriented towards each other.

While this is an important biological problem, unfortunately, the manuscript currently lacks major technical controls, in particular controls for correct spectral separation of the employed fluorophores in the RNA FISH experiments. In addition, figure labels and images lack clarity, making many results difficult to interpret.

Major comments:

- Please provide detailed control experiments for fluorophores used in the RNA FISH experiments, in particular:
 - o a) perform lambda scans for all individual fluorophores to show spectral separation. It is essential that excitation of one fluorophore does not lead to the excitation of another one, which could severely hamper the interpretation of the spatial positioning. For example, FAM and Qasar570 have a significant spectral overlap, and at 495 nm FAM is excited 100% and Q570 is excited 28% (<https://www.chroma.com/spectra-viewer?fluorochromes=10427%2C10433%2C10541%2C10542>), therefore combining these two fluorophores in the RNA FISH experiments might result in false-positive co-localization and might make it impossible to distinguish individual mRNA regions
 - o b) test for chromatic aberrations using multi-color fluorescent beads (e.g. tetraspec beads). If necessary, perform image registration.
 - o a useful guideline for particle detection and image registration can be found in the smFISH/SIM protocol from the Zenklusen laboratory (https://link.springer.com/protocol/10.1007/978-1-0716-0935-4_17)
- Figures should contain individual fluorescent channels to complement the merged channel images, since in the merged images it is often not possible to distinguish the individual signals making it difficult to see spatial relationships (e.g., Figure 5, Figure S5 where, due to the signal overlay, the mRNA particle cannot be distinguished from the NPC signal).
- For the mRNA distance measurements: Please provide data of actual distances between different probes (e.g., using a violin plot) rather than using descriptions such as '5'-3' in proximity' or '5'-3' distant'. See Adivarahan et al. Mol Cell 2018 for examples.
- In all figures and text: provide information about which fluorophore was used for each experiment, for example, instead of writing 'magenta', 'cyan' and 'green' indicate name of fluorophores used.
- Describe how individual channels were assigned to a single mRNA: What is the maximum distance between 2 or 3 colors in order to be assigned to one mRNA particle? What is the frequency of 'lone' signals (only 5' or 3' end), which could refer to active mRNA transcription or degradation? Are particles only counted if they are in a specific signal intensity range, and if so, how are bright sites of transcription excluded from the analysis, where assignment of signal to an individual mRNA is not possible?
- Discuss similarities and differences to Adivarahan et al. Mol Cell 2018, where the same RNA FISH strategy was employed to analyze the spatial organization of single mRNPs in the nucleus, the cytoplasm and in stress granules. In particular, discuss the discrepancy of the spatial organization of cytoplasmic translating mRNAs.
- Provide additional images of the Cerulean-SRSF over-expressions (Figure 2), to confirm that SRSF wild type and mutants are expressed to a similar extent, and to show their localization in the nucleus relative to the individual mRNAs (e.g., to show that the RNAs are not only trapped in SRSF speckles, which could affect their spatial organization due to inter-molecular rearrangements rather than intra-molecular changes in compaction). Ideally, also provide RNA-immunoprecipitations to show differential binding of the wild type and mutant to the GFP-Dys-MS2 transcript, which would provide more direct evidence that SRSF overexpression leads to stronger mRNP binding and therefore directly affects mRNP compaction.
- Figures 5 and S5 show mRNAs that are exported through the NPC with their 5' end first that could be mRNAs that are actively translated by ribosomes (which localize close to the nuclear pore). Treating cells with translation inhibitors such as puromycin should evict ribosomes from translating mRNAs and should block engagement of mRNAs with those peripheral ribosomes and allow for the visualization of only the exported mRNAs.

Minor comments:

- Line 157: write out reference (According to Singh et al. 2012, ...)
- Line 208: wouldn't high levels of mRNA be detrimental for correct assignment of signals to individual mRNPs? What was the cut-off for mRNA abundance?
- Line 210: MKI67 transcripts: provide data for average number of mRNAs per nucleus and per cell

- Line 210: provide sequences for all RNA FISH oligos in methods section
- Line 243: provide a reference for Na-azide
- Line 243: Na-Azide is quite unspecific and also causes formation of stress granules (see e.g. Buchan, Yoon and Parker, JCS 2011), which in turn could affect mRNA compaction, similar to what Adivarahan et al. (Mol Cell 2018) had seen. As an alternative, Antimycin A, which has not been shown to cause SG formation, should be used to determine mRNP compaction state
- Line 248: please show the data
- Line 252: please discuss why CHX causes mRNP collapse, as the drug should lock ribosomes onto the mRNA and therefore not affect spatial mRNP organization, similar to what has been reported by Adivarahan et al. Mol Cell 2018
- Line 273: mention RFP670 tag in methods section
- In discussion: please refer to the individual figures when discussing the respective results
- Line 327: NXF1 recruitment data is discussed but no data is provided, please include the respective figure
- Line 336: how much does is this comparatively small change in mRNP diameter expected to affect particle diffusion? Could also other factors, e.g., increased RBP-DNA interactions, affect diffusion, independently of particle size?
- Line 359: Adivarahan et al. (Mol Cell 2018) have shown that the 5' and 3' ends of cytoplasmic mRNAs are not in close proximity, and these mRNAs might therefore not form a circle. Please include this differential view in the discussion.
- Line 369: '...', cause a larger decrease this association,....': there seems to be a word missing
- Line 564: please indicate number of biological replicates and RNAs per replicate

Figure comments:

- Figure 1:
 - o 1A: The GFP-Dys-MS2 transcript displays a strong signal at the site of transcription, which could indicate impaired transcriptional processing, possibly due to the MS2 stem loops which have previously been shown to affect mRNA processing (Garcia and Parker 2015, 2016; Haimovich et al. 2016). Please provide control FISH data for a GFP-Dys mRNA without the MS2 loops, to check if MS2 confounds mRNP conformation. Alternatively/additionally, including other predominantly nuclear mRNAs (e.g. Prasanth et al. 2005 or Bahar-Halpern et al. 2015) and determining their spatial conformation would significantly strengthen the manuscript.
 - o 1B: individual mRNAs are not visible. Please provide images with enhanced contrast.
 - o 1C: increase the size of the example particles, show individual channels and add a scale bar.
- Figure 2:
 - o 2A: Please show expression and localization of over-expressed Cer-SRSFs
 - o 2B: Why are only 3 statistical tests shown? If the others are not significant, please indicate this in the figure legend.
 - o 2B: How is 'transcription site' defined? Does this exclude zone of strong signal cluster where distinguishing individual mRNAs is not possible?
- Figure 3:
 - o What is the difference between B and D? Merge?
 - o 4C: please provide example images of nuclear transcripts, similar to Figure 4D.
- Figure 4:
 - o 4A: rod-like shape, e.g., in the CHX treatment, is very hard to see in example mRNAs, they rather look like collapsed mRNAs. Please provide alternative images.
- Figure 5:
 - o 5A and B: please provide individual images of all channels, because overlay of mRNA signals hides POM121 signal
 - o 5B: please also provide example images for TUG1 and NORAD
- Figure S1:
 - o S1B: please provide an example overview image of a full cell in addition to the schematic
 - o S1D: please provide x- and y-axes for the graphs
- Figure S2:

- o S2D: it is not clear from the images that this is MKI67 instead of GFP-Dys-MS2. Please add a label or provide a schematic similar to Figure S2A.
- Figure S3:
 - o Does lamin A knock-out affect gene expression and overall mRNA abundance as well as gene/transcript localization in the nucleus? Please provide RNA images for individual conditions and mRNA quantifications
 - o Provide DAPI image to show state of chromatin compaction
 - o S3B: scale bar is missing
 - o S3C: Control experiment missing: determine mRNP organization in U2OS + hyper-osmotic shock (in Lamin A wild type condition)
- Figure S4:
 - o Please provide scale bar for zoom-ins
- Figure S5:
 - o Please provide individual channels for all images, it is very difficult to see individual pores
 - o Please provide scale bar for zoom-ins
- Figure S7:
 - o Please provide individual channels for all images, it is very difficult to see individual pores

We thank the reviewers for their insightful comments and suggestions. We have made a large effort to address the issues raised, specifically the technical details as mentioned. We were happy to learn that the reviewers found significant importance in this study and hope they will feel, as we do, that the manuscript has substantially improved in light of their remarks.

We added new data in the figures and supplementary figures. Issues which we thought did not necessarily need to be part of the supplementary figures were prepared specifically for the reviewers and appear at the end of this file.

Reviewer #1:

In the abstract the authors state that: “It is assumed that at the nuclear pore complex (NPC), the mRNP unfolds and enters 5'-first into the pore.” In the introduction the authors nicely underline the notion that the 5' end enters first with the relevant literature. But why is it intuitive that to assume the mRNP has to unfold? The authors state themselves that Balbini ring mRNPs are globular during export. Furthermore, FG-Nups and mRNPs have different biophysical properties and thus likely need to minimize their interaction surface.

Indeed, it is unclear what shape mRNPs undertake during export. The BR RNPs are very large with a globular structure during travel in the nucleoplasm but the EM images show them restructuring and taking on a ribbon-like shape when passing through the pore, probably since they are too large to fit in the channel of the pore (PMID: 1586943). We too saw this when looking at large mRNPs (PMID: 20453848). For the case of smaller RNAs and smaller RNPs we agree that these would probably not always need to unfold. The text in the Introduction has been refined accordingly.

Line 116: May the massive MS tagging at the 3' end interfere with how the mRNP enters the NPC?-

In a previous study we described the generation of this transcript and the export of the GFP-Dys-MS2 mRNA using live-cell imaging and could also detect the exit of the mRNA from the NPC in real-time (PMID: 20453848). mRNAs could be detected in the cytoplasm as well as the protein that is translated from these transcripts, which is fluorescent and detectable. Many studies in mammalian cells have added the MS2 tag to the 3'UTR. There haven't been significant effects documented for export. In the current study, we used this particular transcript only to study the compaction levels of the nuclear RNAs. In light of the finding herein that the 5' is usually first we can assume that the MS2 on the 3'-end has less of an effect.

Fig 1B: You should quantify this data.

We appreciate this suggestion. We quantified the data, now added as a supplementary figure S1. The quantifications were on par with our observations, and show the rate at which the polymerase passes through the beginning and end of the gene.

Fig. S1D: What exactly is the difference between left middle and right panel? Different positions in the nucleus? This should be labeled more clearly. On a general note, the authors often place explanations into the main text that alternatively could be into the figure legends.

This was indeed unclear. The figure shows examples of several mRNAs with different degrees of compaction for the GFP-Dys-MS2 mRNAs we analyzed in the nucleus. The figure was refined to explain its parts more clearly. In addition, we edited the text and tried to move information to the legends.

Fig 2C: The plot indicates a rather large spread of the data but in this representation its hard to tell. You may choose violin plots or something that allows a more objective judgment.

We thank the reviewer for this suggestion and changed all box plots to violin plots.

Fig. 3C: Can one really conclude circularity? Why exactly? Do you really mean physical circularity? The authors somehow would have to show how this differs from random non-circular conformations, e.g. by comparing to a cleaved or otherwise non-circular transcript. This could be addressed by re-phrasing more accurately. You may also compare to data from the next section of the paper (cycloheximide treatment) to address this. Statistical tests would be good.

We agree that we cannot conclude circularity. That is why in the figure we used the term 5'-3' ends in proximity. Also, the signals do not overlap so proximity is a preferred term. The choice of wording in the text should have been better and we now have rephrased in several places in the text as suggested.

Line 274: I suggest to delete 'towards the nucleoplasmic side', unless you have more clear evidence. Please note that Nup62 is a member of both the Nup214 and Nup62 complexes. You see POM121 between basket and CF Nups. That's good enough.

We corrected as suggested.

Line 287: You are probably right, but 1st time claims are tricky and may upset people. Nobody can be aware of all literature that has been published.

We removed this statement.

Line 302/ 419: 'Gene gating' would mean that it is also transcribed at the nuclear periphery, right? Strictly speaking, you have not shown that. There could be alternative explanations (defined tracks towards the export site, splitting of particles close to NPCs or coincidence which would bring us back to statistical tests). You could down town and say that local clustering of exported transcripts may be indicative of gene gating. Or repeat the assay shown in Figure 1B with that particular mRNA and show that its close by.

We agree and think that this transcript is transcribed close to the nuclear periphery since it is situated on chromosome 10 which tends to be peripheral. We did not manage to clearly detect

the transcription sites of this mRNA so we generated probes to the intron, however, we were not able to confidently say that we detect the sites of transcription, even though the transcripts were abundant. In addition, as suggested by the reviewer, we added an experiment with another transcript that we know is transcribed close to the periphery – this is the lncRNA transcript CCAT1, transcribed from chromosome 8, which is also a peripheral chromosome. We find that many of the CCAT1 transcripts export through NPCs close to the site of transcription (Figure S13B). We modified the text to say this might be indicative of gene gating.

Reviewer #2

MAJOR COMMENTS

1) When distances between probes on the same molecule are measured, resolution of probes of different colours is critical. Resolution is one of the most spectacular advance in this study. Thus it should be strongly supported at least supplementary data. In the material and methods section, the authors say they estimate the chromatic distortion by labelling repeated sequences with probes of different colours. The data used to determine a 7 nM average deviation, should appear and detailed in Supplementary figures. How many images were analysed? Show the distribution. Is the chromatic shift homogeneously centred around a median of 0 within the entire observation field?

We thank the reviewer for these suggestions and added a supplementary figure (Fig. S2) consisting of controls as an example of measurements of two different fluorescent probes to the same MS2 sequences, along with the distribution of these MS2-MS2 distances. The deviation was determined by the average MS2-MS2 distance across all transcripts, and the median distance was 0 nm. We analyzed 290 transcripts in the GFP-Dys-MS2 expressing cells over 3 repeats. While deviation does exist, it is minor, and the large sample size (hundreds of transcripts) should dampen any measurement bias. We also calibrated the measurements using tetra-fluorescent beads (Fig. L1 for the reviewer), establishing no pixel shift. The deviation is now 9 nM (new measurements were added, >200 transcripts).

2) Is the 7 nm resolution in x,y? What is the resolution in z?

Yes, this resolution is in XY. The Z resolution is ~200 nm.

3) How are distances between probes determined? Did you take into account that mRNAs are 3D objects? A vertical rod might appear as a “compact” object.

The distances were measured using the LASX program, by Leica. The comment by the reviewer is in place and we should have explained it - for the mRNA organization analysis, we only characterized transcripts where the 3 tags were separately identifiable. In the analyzed transcripts, the size of the dots had to be identical, so that they were found within the same Z plane (which has a resolution of 200 nm as mentioned above, and so we only used those transcripts that are fully in the XY plane). It is always possible that some transcripts were

misrepresented, but the large number of transcripts analyzed should insure a more exact measurement. This is now mentioned in the Methods section.

4) Histograms in Figure 3C, 3E and 4B lack statistical analysis of distances (i.e.. box plots) and discussion of the cut-off distance between 5'-3' distant and 5'-3' in proximity classes. How many images cannot be ascribed to one or the other class?

Following suggestions from the reviewers, we added a supplementary figure (Fig. S6), demonstrating compaction analysis depicting cytoplasmic mRNP compaction, adding a statistical dimension to mRNP organization. The data from these measurements also now shows that translation inhibition treatments, such as Puromycin or Cycloheximide, affect the compaction of endogenous mRNA transcripts, making them more compact compared to transcripts in WT cells. Much like the mRNA organization experiments, the effect was reversible upon a 1 hour of wash with medium, and the transcripts returned to a rather open conformation. The Puromycin data is in line with other studies using Puromycin and measuring compaction (Adivarahan et al., 2018), and we add more data regarding Cycloheximide translation inhibition and the flexibility of the transcripts.

5) Show data proving that protein synthesis resumes after removal of cycloheximide (Figure 4).

We conducted several experiments to make sure that protein synthesis is inhibited upon treatment, and resumes upon wash. We prepared a figure for the reviewer (Fig. L2) showing the appearance of fluorescently-tagged peroxisomes in cells expressed from an inducible gene (PMID: 21264352), representing that general protein expression is active. We demonstrate how peroxisomes are abundant in the cytoplasm of activated cells, disappear upon cycloheximide treatment, and return after a 1-hour wash.

6) Is RNA export orientation (Figure 5) sensitive to protein inhibition (cycloheximide) to address a possible contribution of ribosomes to export orientation?

This is an interesting question which we went and examined. We added Fig. S12 exploring the degree of directionality upon translation inhibition, using both Puromycin and Cycloheximide. The results show no effect on export directionality compared to transcripts in untreated cells, suggesting that the state of translation does not affect mRNA export directionality. This would also concur with the other data showing that lncRNAs also exit 5'-end first even though they are not translated.

7) Other labs using different RNAs made different observations for the impact of protein synthesis on mRNA compaction. Is the discrepancy due to the RNAs investigated? One should probe at least one of the previously investigated mRNAs to discriminate between a technical effect or target effect.

We repeated the experiment performed in (Adivarahan et al., 2018) with the MDN1 transcript. We prepared the same probes sets used in that study and basically obtained very similar effects (see Figure L7 for the reviewer). In addition, in light of the comments, we examined these

effects on another mRNA, NCOA3, and found that it too becomes more compact following translation inhibition, same as MKI67 (see Figure S7).

MINOR COMMENTS:

Numbers such as 42.892 nm (Fig S1), 7.128 (methods) are meaningless. 43 nm and 7 are more appropriate.

Accepted and corrected.

Reviewer #3:

Major comments:

- Please provide detailed control experiments for fluorophores used in the RNA FISH experiments, in particular:

a) perform lambda scans for all individual fluorophores to show spectral separation. It is essential that excitation of one fluorophore does not lead to the excitation of another one, which could severely hamper the interpretation of the spatial positioning. For example, FAM and Qasar570 have a significant spectral overlap, and at 495 nm FAM is excited 100% and Q570 is excited 28% (<https://www.chroma.com/spectra-viewer?fluorochromes=10427%2C10433%2C10541%2C10542>), therefore combining these two fluorophores in the RNA FISH experiments might result in false-positive co-localization and might make it impossible to distinguish individual mRNA regions-

This is an important point and indeed we tested the imaging sequence to examine if we have any leaks between the channels using the featured fluorophores. The results are presented in the figure for the reviewer (Fig. L3). This figure shows the merge of 3 channels in one image, and only one signal is detected. We used one fluorophore at a time, imaging with all 3 channels to examine for any fluorescent leaks. While there was some “dirt” in some of the images, it could not be mistaken for transcripts, and we did not detect any leaks between the channels.

b) test for chromatic aberrations using multi-color fluorescent beads (e.g. tetraspec beads). If necessary, perform image registration.

An aberrations test was done once a month when using the STED. We attach a figure for the reviewer (Fig. L1) showing the use of tetra-fluorescent beads. The test shows perfect overlap between all channels, ruling out the possibility of chromatic aberrations or shifts.

- Figures should contain individual fluorescent channels to complement the merged channel images, since in the merged images it is often not possible to distinguish the individual signals

making it difficult to see spatial relationships (e.g., Figure 5, Figure S5 where, due to the signal overlay, the mRNA particle cannot be distinguished from the NPC signal).

We provide the individual channels now in figures S9, S11, S12.

- For the mRNA distance measurements: Please provide data of actual distances between different probes (e.g., using a violin plot) rather than using descriptions such as '5'-3' in proximity' or '5'-3' distant'. See Adivarahan et al. Mol Cell 2018 for examples.

As per the reviewers' recommendations, we changed all box plots to violin plots, in order to better demonstrate the distribution of the data, and distances are noted where relevant. We prefer to also include the simplistic histograms that provide a simple but informative yes/no answer to the question.

- In all figures and text: provide information about which fluorophore was used for each experiment, for example, instead of writing 'magenta', 'cyan' and 'green' indicate name of fluorophores used.

Text and figures were refined.

- Describe how individual channels were assigned to a single mRNA: What is the maximum distance between 2 or 3 colors in order to be assigned to one mRNA particle?

The maximum distance measured was approximately 300 nm, provided that there were no transcripts adjacent to it, so that we can conclude that both labels are on the same transcript. This was added to the methods.

What is the frequency of 'lone' signals (only 5' or 3' end), which could refer to active mRNA transcription or degradation?

We performed analysis of 228 triple-tagged cytoplasmic MKI67 transcripts in U2OS cells (Fig. L4 for the reviewer), showing that 13% exhibited a lone signal, 21% exhibited a double signal, and the majority (66%) of the transcripts demonstrated a triple signal.

Are particles only counted if they are in a specific signal intensity range, and if so, how are bright sites of transcription excluded from the analysis, where assignment of signal to an individual mRNA is not possible?

For analysis, the particles were counted if their size (a signal of 0.2 microns) and intensity (1500-1000 units, LASX) were within range, and being much brighter than this threshold, transcription sites were excluded from analysis. This was added to the methods.

- Discuss similarities and differences to Adivarahan et al. Mol Cell 2018, where the same RNA FISH strategy was employed to analyze the spatial organization of single mRNPs in the nucleus, the cytoplasm and in stress granules. In particular, discuss the discrepancy of the spatial organization of cytoplasmic translating mRNAs.

We added another part in the discussion regarding this landmark study. We used a very similar system to Adivarahan et al. Mol Cell 2018, implementing smFISH and high-resolution microscopy techniques. We did study different transcripts, but found that they act the same upon translation inhibition, namely, all 3 transcripts measured by us were more dense under treatment, similar to what the former study found. We also performed the experiments on the MDN1 transcript used by Adivarahan et al and we come to the same conclusions regarding conditions of translation inhibition. When examining mRNA organization, we took different approaches; while the former study used 3D imaging and analysis, we chose to focus on the 2D images. The cytoplasmic transcripts we examined exhibited 5'-3' proximity which grew smaller in response to translation inhibition. This could be due to the varied measurement and analysis techniques, or it could imply that not all mRNAs act the same. Further research is needed in this area.

- Provide additional images of the Cerulean-SRSF over-expressions (Figure 2), to confirm that SRSF wild type and mutants are expressed to a similar extent, and to show their localization in the nucleus relative to the individual mRNAs (e.g., to show that the RNAs are not only trapped in SRSF speckles, which could affect their spatial organization due to inter-molecular rearrangements rather than intra-molecular changes in compaction).

Attached is a figure for the reviewer (Fig. L5) showing the intensity of different SRSFs. We demonstrate that the intensity between the overexpressed proteins is rather similar (Fig. L5A), and that the vast majority of GFP-Dys-MS2 transcripts are not found inside speckles (Fig. L5B), as expected for mRNAs. In fact, only a tiny fraction of the transcripts was found in nuclear speckles, meaning the analysis was unaffected by speckle enlargement due to SRSF overexpression.

Ideally, also provide RNA-immunoprecipitations to show differential binding of the wild type and mutant to the GFP-Dys-MS2 transcript, which would provide more direct evidence that SRSF overexpression leads to stronger mRNP binding and therefore directly affects mRNP compaction.

This experiment requires generating sets of stable cell lines but this has turned out to be unexpectedly challenging and while we did attempt to comply, in spite of several attempts we have not managed to make proper headway with the establishment of these lines. However, we do rely upon the work of Singh et al, showing enhanced SRSF1 binding to mRNA transcripts.

- Figures 5 and S5 show mRNAs that are exported through the NPC with their 5' end first that could be mRNAs that are actively translated by ribosomes (which localize close to the nuclear pore). Treating cells with translation inhibitors such as puromycin should evict ribosomes from translating mRNAs and should block engagement of mRNAs with those peripheral ribosomes and allow for the visualization of only the exported mRNAs.

This is an interesting question which we went and examined. We added Fig. S12 exploring the degree of directionality upon translation inhibition, using both Puromycin and Cycloheximide. The results show no effect on export directionality compared to transcripts in untreated cells,

suggesting that the state of translation does not affect mRNA export directionality. This would also concur with the other data showing that lncRNAs also exit 5'-end first even though they are not translated.

Minor comments:

- Line 157: write out reference (According to Singh et al. 2012, ...)

Added.

- Line 208: wouldn't high levels of mRNA be detrimental for correct assignment of signals to individual mRNPs? What was the cut-off for mRNA abundance?

While the expression of GFP-Dys-MS2 could be high enough to fill much of the nucleus, transcripts which were too close to other transcripts were not analyzed (since we were unable to distinguish between the individual ones), to avoid bias. This was added to the methods.

- Line 210: MKI67 transcripts: provide data for average number of mRNAs per nucleus and per cell.

Imaris analysis yielded the following numbers: 61 transcripts per nucleus, 234 transcripts per cell were measured. We added the number of transcripts in the nucleus and cytoplasm for all transcripts in the legend of Fig. 1A.

- Line 210: provide sequences for all RNA FISH oligos in methods section.

We added a comprehensive table of all probe sequences in use.

- Line 243: provide a reference for Na-azide.

Added.

- Line 243: Na-Azide is quite unspecific and also causes formation of stress granules (see e.g. Buchan, Yoon and Parker, JCS 2011), which in turn could affect mRNA compaction, similar to what Adivarahan et al. (Mol Cell 2018) had seen. As an alternative, Antimycin A, which has not been shown to cause SG formation, should be used to determine mRNP compaction state.

We appreciate this comment, and conducted Antimycin A experiments, characterizing cytoplasmic mRNA organization in untreated cells compared to Antimycin-treated cells (Fig. L10 for the reviewer). We found that, similar to other ATP depletion treatments, the transcripts in cells subjected to Antimycin treatment tended to exhibit a more 5-3 distant form compared to transcripts in untreated cells.

- Line 248: please show the data.

The Na-Azide treatment was insignificant. We removed said sentence and refined the text.

- Line 252: please discuss why CHX causes mRNP collapse, as the drug should lock ribosomes onto the mRNA and therefore not affect spatial mRNP organization, similar to what has been reported by Adivarahan et al. Mol Cell 2018.

We added a part in the discussion.

- Line 273: mention RFP670 tag in methods section.

Added.

- In discussion: please refer to the individual figures when discussing the respective results.

We appreciate this constructive comment, and refined the text to point towards figures.

- Line 327: NXF1 recruitment data is discussed but no data is provided, please include the respective figure.

We prepared Fig. L6 for the reviewer. This is a recruitment experiment showing that while some factors, such as Aly/REF, are co-transcriptionally recruited to the GFP-Dys-MS2 site of transcription, NXF1 does not bind to the site.

- Line 336: how much does is this comparatively small change in mRNP diameter expected to affect particle diffusion? Could also other factors, e.g., increased RBP-DNA interactions, affect diffusion, independently of particle size?

The diffusion coefficient is affected by the diameter of the particle. This has been discussed in the supplementary part of the original paper following mRNPs diffusion (PMID 15205532). It is plausible that other factors and conditions could affect particle diffusion, for instance additional protein might cause increased non-specific interactions with structures/obstacles during the movement of the mRNP in the nucleoplasm. These are all interesting points but they have not been tested experimentally so far in the mammalian nucleus.

- Line 359: Adivarahan et al. (Mol Cell 2018) have shown that the 5' and 3' ends of cytoplasmic mRNAs are not in close proximity, and these mRNAs might therefore not form a circle. Please include this differential view in the discussion.

Added to discussion.

- Line 369: '...', cause a larger decrease this association,....': there seems to be a word missing.

Refined.

- Line 564: please indicate number of biological replicates and RNAs per replicate.

Added to statistical analysis in the methods section.

Figure comments:

- Figure 1:

1A: The GFP-Dys-MS2 transcript displays a strong signal at the site of transcription, which could indicate impaired transcriptional processing, possibly due to the MS2 stem loops which have previously been shown to affect mRNA processing (Garcia and Parker 2015, 2016; Haimovich et al. 2016). Please provide control FISH data for a GFP-Dys mRNA without the MS2 loops, to check if MS2 confounds mRNP conformation. Alternatively/additionally, including other predominantly nuclear mRNAs (e.g. Prasanth et al. 2005 or Bahar-Halpern et al. 2015) and determining their spatial conformation would significantly strengthen the manuscript.

We expressed a the GFP-Dys plasmid that does not have MS2 repeats and performed the FISH experiment with the probe to the 5'-end and the middle of the gene, as done for the GFP-Dys-MS2 mRNA (Fig. L11 for the reviewer), and found no significance difference in the compaction and distribution compared to the MS2-containing plasmid.

1B. individual mRNAs are not visible. Please provide images with enhanced contrast.

The focus of this image was the transcription site and the proof of concept that we can separately detect the 5' and 3' signals, and so single transcripts were not enhanced in this image. To improve the presentation of this experiment, we added an analysis of transcription site sizes at different activation times (Figs. S1 and S3).

1C: increase the size of the example particles, show individual channels and add a scale bar. Found at Fig. S3.

- Figure 2:

2A: Please show expression and localization of over-expressed Cer-SRSFs.

This appears in the figure for reviewer - Fig. L5.

2B: Why are only 3 statistical tests shown? If the others are not significant, please indicate this in the figure legend.

The other tests were not significant, and were added to the legend.

2B: How is 'transcription site' defined? Does this exclude zone of strong signal cluster where distinguishing individual mRNAs is not possible?

A transcription site is the strong and distinguishable zone where one cannot differentiate between single transcripts.

- Figure 3: What is the difference between B and D? Merge?

Panel B is an example of transcripts found in a single cytoplasm, while panel D features transcripts from several cells. This is now mentioned in the legend.

o 4C: please provide example images of nuclear transcripts, similar to Figure 4D.

Images were added to figure S4D.

Figure 4: 4A: rod-like shape, e.g., in the CHX treatment, is very hard to see in example mRNAs, they rather look like collapsed mRNAs. Please provide alternative images.

As per the reviewer's recommendations, we added a figure (Fig. S6) exploring MKI67 compaction under translation inhibition conditions. As new analysis shows, the transcripts are indeed more collapsed, hence the technical difficulty to get better images.

-Figure 5: 5A and B: please provide individual images of all channels, because overlay of mRNA signals hides POM121 signal.

Added as Fig. S9.

5B: please also provide example images for TUG1 and NORAD.

Images of TUG1 and NORAD export out of the NPC are presented in Fig. S11.

- Figure S1:

S1B: please provide an example overview image of a full cell in addition to the schematic.

This is presented in Fig 1A, right hand image.

S1D: please provide x- and y-axes for the graphs.

This information was added to graphs.

- Figure S2: S2D: it is not clear from the images that this is MKI67 instead of GFP-Dys-MS2. Please add a label or provide a schematic similar to Figure S2A.

We changed Figure S4 and added an explanation in the legend.

- Figure S3: Does lamin A knock-out affect gene expression and overall mRNA abundance as well as gene/ transcript localization in the nucleus? Please provide RNA images for individual conditions and mRNA quantifications.

Following this comment, we performed a quantification of MKI67 transcripts in WT and LMNA-KO cells and show this with the images in Fig. L8 for the reviewers.

o Provide DAPI image to show state of chromatin compaction. Added.

o S3B: scale bar is missing. Added.

o S3C: Control experiment missing: determine mRNP organization in U2OS + hyper-osmotic shock (in Lamin A wild type condition).

The hyper-osmotic treatment on U2OS WT cells yielded no significant change in compaction. We added to text (and Fig L9 for the reviewers).

- Figure S4: Please provide scale bar for zoom-ins. Figure S5: Please provide individual channels for all images, it is very difficult to see individual pores. Figure S7: Please provide individual channels for all images, it is very difficult to see individual pores.

We now provide the separate channels and scale bars.

Figures for the reviewers:

Figure L1: Tetra-fluorescent beads aberration test. Tetra-fluorescent beads were imaged using our imaging sequences and in different wavelengths, and measured in the LASX program to test for overlap.

Figure L2: Protein synthesis resumes after removal of cycloheximide. CFP-peroxisomes (cyan) are induced by doxycycline (dox) in Tet-On U2OS cells. Their appearance is dependent on active translation and disappear when cells are treated with cycloheximide (CHX). They reappear after washing out of the CHX. The scheme (left) show when the different compounds were added. DNA stain (magenta). Scale bar, 10 μ m.

Figure L3: Fluorescent leak verification. MKI67 transcripts in U2OS cells were labeled using one probe set at a time (Left- 670 nm, middle- 570 nm, right- 520 nm) and imaged in 4 channels (DAPI, GFP, Cy3 and Cy5) using the typical imaging sequence. Scale bar, 10 μ m.

Figure L4: Lone signal frequency. Cytoplasmic MKI67 transcripts in U2OS cells were tagged using 3 fluorophores, and analysis was done to determine how many of them were actually single, double or triple tagged.

Figure L5: SRSFs intensity and mRNP distribution in relation to nuclear speckles. (A) Expression levels of Cerulean (control), Cer-SRSF1, Cer-SRSF4 and Cer-SRSF1-no-RS, at the same exposures. Scale bar, 10 μm . (B) Localization of GFP-Dys-MS2 transcripts (magenta, 570 nm) in relation to Cer-SRSF4 (cyan). Scale bar, 10 μm .

Figure L6: NXF1 is not recruited to the GFP-Dys-MS2 transcription site. RNA FISH to MS2 sequences show the active GFP-Dys-MS2 transcription site (magenta, 570 nm) along with immunofluorescence to NXF1 (orange) and Aly/REF IF (green). Only Aly-REF is recruited co-transcriptionally. Scale bar, 10 μm .

Figure L7: MDN1 compaction changes under translation inhibition conditions. (A) Double tagging of MDN1 mRNA transcripts in untreated, Puromycin and Cycloheximide-treated U2OS cells. Middle of the transcript was tagged using probes at 670 nm (cyan) and the 5'-end was tagged using probes at 570 nm (magenta). Right – enlarged examples of single mRNAs. Scale bars, 8 μ m and 0.5 μ m. (B) Violin plots comparing MDN1 mRNA 5'-mid distance distribution for untreated, Puromycin and Cycloheximide-treated U2OS cells. NT n=284 transcripts, Puro n=446 transcripts, Cyclo n=164 transcripts. ****p<0.0001. Boxplots show the distance (nm) (center line, median; box limits, upper and lower quartiles; whiskers, minimum to maximum range; dots, outliers).

Figure L8: MKI67 expression in U2OS and *LMNA*-KO cells. (A) MKI67 transcripts (white) in WT U2OS cells (left) compared to U2OS LMNA-KO cells (right). Scale bar, 10 μ m. (B) MKI67 Nucleus/Cytoplasm ratio in WT and LMNA-KO U2OS cells, WT n= 1092 transcripts, LMN-KO n=2045 transcripts.

Figure L9: Hyper-osmolarity has no significant effect on mRNP compaction in WT U2OS cells. 5'-middle distances of nuclear MKI67 transcripts under hyper-osmolar conditions in U2OS cells. U2OS n=171, U2OS hyper-osmolar conditions n=206 transcripts. $p < 0.005$. Boxplots show the distances (nm); center line, median; box limits, upper and lower quartiles; whiskers, minimum to maximum range; dots, outliers.

Figure L10: Antimycin A treatment affects mRNA organization. The form of cytoplasmic MKI67 transcripts under Antimycin A conditions in U2OS cells. U2OS No treatment n=124, U2OS Antimycin A conditions n=47 transcripts.

Figure L11: GFP-Dys without MS2. Double tagging of Full-Dys mRNA transcripts (A) without MS2 sequences in the 3'UTR and (B) with MS2 sequences. 5'-end of the transcript was tagged using probes at

488 nm (green) and the middle was tagged using probes at 570 nm (magenta). Right – enlarged examples of single mRNAs. Scale bars, 10 μm and 0.5 μm . (B) Violin plots comparing mRNP compaction of GFP-Dys and GFP-Dys-MS2 transcripts. Boxplots show that the relative distances were not significantly different (center line, median; box limits, upper and lower quartiles; whiskers, minimum to maximum range; dots, outliers).

REVIEWERS' COMMENTS

Reviewer #1 (Remarks to the Author):

The authors have addressed all of my comments. I most enthusiastically recommend this manuscript for publication!

Reviewer #2 (Remarks to the Author):

In my view the authors have correctly addressed the reviewers comments. I strongly support publication of this important piece of work.

Reviewer #3 (Remarks to the Author):

The authors have done a good job addressing our concerns, and we now recommend publication.